# SENSITIVITY VERIFICATION FOR ADDITIVE DECISION TREE ENSEMBLES

**Arhaan Ahmad, Tanay V. Tayal, Ashutosh Gupta & S. Akshay**
Department of Computer Science and Engineering,
Indian Institute of Technology Bombay,
Mumbai, India.
`{arhaan, tanaytayal, akg, akshayss}@cse.iitb.ac.in`

## ABSTRACT

Tree ensemble models, such as Gradient Boosted Decision Trees (GBDTs) and random forests, are widely popular models for a variety of machine learning tasks. The power of these models comes from the ensemble of decision trees, which makes analysis of such models significantly harder than for single trees. As a result, recent work has focused on developing exact and approximate techniques for questions such as robustness verification, fairness and explainability for such models of tree ensembles.

In this paper, we focus on a specific problem of feature sensitivity for additive decision tree ensembles and build a formal verification framework for it, where we also take into account the confidence of the tree ensemble in its output. We start by showing theoretical (NP-)hardness of the problem and explain how it relates to other verification problems. Next, we provide a novel encoding of the problem using pseudo-Boolean constraints. Based on this encoding, we develop a tunable algorithm to perform sensitivity analysis, which can trade off precision for running time. We implement our algorithm and study its performance on a suite of GBDT benchmarks from the literature. Our experiments show the practical utility of our approach and its improved performance compared to existing approaches.

## 1 INTRODUCTION

Tree ensemble models, such as gradient boosted decision trees (Friedman, 2001) and random forests (Breiman, 2001), are now widely used for machine learning tasks in domains ranging from banking applications (Madaan et al., 2021) to computer vision (Criminisi & Shotton, 2013) to transportation (Podgorelec et al., 2002). The power of these models comes from the ensembling or boosting, which is known to empirically improve performance, unlike single decision trees, which are explainable and simple to understand but unwieldy to model complex behaviour. XGBoost (Chen & Guestrin, 2016), one such popular tree ensemble learning algorithm, shows remarkable performance with tree ensembles of size 100, where each tree has depth at most 5 or 6. Of course, the tradeoff with such modelling power is that it becomes difficult to predict and analyze these models, i.e., to ensure that they are reliable, robust, and behave as expected. This has led to a rich line of work in the last decade on formalizing and verifying different properties of tree ensembles.

The first such property is robustness checking which asks whether there are adversarial input perturbations that could lead to misclassification. This problem has been addressed by several works over the past 5 years including Chen et al. (2019b); Einziger et al. (2019); Devos et al. (2021), using different techniques ranging from optimization/MILP-based (Kantchelian et al., 2016) to SMT-solver based approaches (Ignatiev et al., 2020a). Using SMT-solvers allows one to give guarantees of soundness and completeness, which is highly desirable when dealing with reliability issues but is often less scalable than purely optimization-based approaches. The literature also distinguishes between local (checking robustness around a given input) and global robustness (checking for all inputs) where the universal quantifier makes the latter problem significantly harder (Chen et al., 2019b; Leino et al., 2021). Many techniques also apply approximations (Devos et al., 2021) or look at subclasses (Andriushchenko & Hein, 2019). Indeed, this is unsurprising since in Kantchelian et al. (2016), it was shown that a very simple related problem is already NP-hard.

Another property of interest is the sensitivity of a subset of inputs or features. A model is sensitive to a set of features if by keeping other features fixed and changing those features, the output of the model changes. In practice, models which are sensitive to a very small subset of features are prone to adversarial attacks. This question has also been seen as a way to formalize fairness (e.g., Dwork et al. (2012)) or causal discrimination (e.g, Galhotra et al. (2017)): sensitive features can be seen as protected, and changing the decision based only on these features can be seen as being unfair. This formulation was also adopted in Calzavara et al. (2023), where an approximation algorithm was provided for the same problem. Further, the generic tool Devos et al. (2021) solves verification questions including sensitivity/fairness, but provides approximate answers using optimization. A formal methods approach towards the problem is also provided in Ignatiev et al. (2020a), where the authors show how the fairness problem can be modelled as a global robustness query and hence relate the above problems. Another related approach in Törnblom & Nadjm-Tehrani (2020) used abstract interpretation techniques for verifying tree ensembles.

In this paper, we launch a deeper investigation into the sensitivity problem for decision tree ensemble models, both from theoretical and practical perspectives. We start by observing that in most models and benchmarks, we are only interested in checking the sensitivity of a fixed few or even just one input feature. So the first question we ask is whether the problem remains hard even for a bounded number of features being changed. Second, if we look at tree ensemble training algorithms, like XGBoost, we observe that the decisions done at leaves, though binary, are derived from real number values, obtained from confidence. This leads us to ask if we can use these numbers as parameters to obtain a parametric notion of sensitivity, which can quantify "how sensitive features are" instead of just saying if they are sensitive or not. Third, we observe that existing encodings of the sensitivity problem either use one extreme of Boolean reasoning (e.g, SMT solvers in Ignatiev et al. (2020a)) or the other extreme of purely optimization-based reasoning (e.g, MILP solvers in Kantchelian et al. (2016)). Can we use other forms of powerful reasoning, such as pseudo-Boolean solvers, that have shown to be effective in other problems (Mexi et al., 2023) for the sensitivity problem? This forms the third question that we try to address in this work.

Surprisingly, we show that even for a single feature, the sensitivity problem remains NP-hard. To show this, we provide a novel reduction from 3CNF-SAT which in fact shows that the problem is NP-hard even when restricted to decision tree ensembles with trees of depth at most 3. Next, we introduce a threshold parameter $p$ to define the class change confidence for the sensitivity problem. In other words, if the output of the classifier is above $p$ threshold, we say that the decision is 1 and if the output the classifier is below $1 - p$, we say that decision is 0. This gives a natural tunable version of the sensitivity problem. Using this, we formulate the sensitivity problem as pseudo-Boolean constraints. Our novelty in the encoding includes a new encoding of trees as pseudo-Boolean constraints, which is rather different from the encoding of trees done in Ignatiev et al. (2020a) for the robustness verification problem. Using this novel encoding we model our property so that we can take advantage of recent advances in pseudo-Boolean solving towards this problem. We then perform experiments to illustrate that our algorithm has an order of magnitude better performance than the state-of-the-art. In sum, our contributions are the following:

1. We formulate a parametric and bounded version sensitivity problem for additive decision tree ensembles and show that it is NP-complete.

2. We develop a novel encoding of this problem using pseudo-Boolean constraints.

3. We implement our algorithm and show that it outperforms state-of-the-art publicly available tools in a suite of benchmarks.

We highlight that our hardness results and encoding ideas work for any tree ensemble which aggregates the trees by summing up the individual tree outputs (these are sometimes called additive tree ensembles, see e.g., Devos et al. (2021)). Such models include GBDTs and random forests which follow an additive predictive model. In particular, our results are independent of the way by which the tree ensemble was trained (e.g, gradient boosting, etc.).

**Other Related Work** In addition to the work mentioned above, there are a few other lines of related work. SMT solvers and solver-based approaches and non-trivial encodings have been widely used for certifying robustness in neural networks and detailed frameworks such as DeepPoly (Singh et al., 2019) and $\alpha, \beta$-Crown (Zhang et al., 2018; 2022) and some of these could also be related

to sensitivity verification. However, there is comparatively less work on using these approaches for decision tree ensembles, beyond the works mentioned in the introduction. Another rich line of related work is to train robust tree ensembles. Calzavara et al. (2020), for instance, trains trees to make them more evasion aware while Chen et al. (2019a) provides a training method that makes the trees more robust. Our current focus is on verifying sensitivity but it would also be interesting to extend this to training insensitive models of tree ensembles. Finally, there is a relation between notions of explainability such as contrastive explainability and abductive explainability (Ignatiev et al., 2020b) and sensitivity or fairness. However, there is a marked difference as we focus on global sensitivity based on given features, while explainability refers to the change of class with respect to or around an input. However, global notions of contrastive explainability could perhaps be related to our approach. As explained above, our approach works for random forests as well. A caveat is that the classical definition of random forests (Breiman, 2001) computes the final answer by max-pooling rather than summing up. We believe that our theoretical results and encodings can also be extended to the max-pool setting, but we leave this for future work. On the other hand, some famous implementations, such as Scikit-Learn (Pedregosa et al., 2011), do use a weighted average of the individual tree predictions to make the final decisions, which can be incorporated into our approach.

## 2 PRELIMINARIES

In the classification setting, given an input space $\mathcal{X} \subseteq \mathbb{R}^d$ defined over $d$-dimensional space of features $\mathcal{F}$, and an output space $\mathcal{Y}$, which is a discrete subset of $\mathbb{R}$, there exists a unknown function $h : \mathcal{X} \to \mathcal{Y}$ that maps each element of $\mathcal{X}$ to its corresponding correct output in $\mathcal{Y}$. A classifier is a function from $\mathcal{X}$ to $\mathcal{Y}$ which approximates $h$ by learning from data. Decision trees and tree ensembles are well-known models used to define classifiers.

**Decision Trees**  A decision tree $T$ is either a leaf $n$ with label $n.val \in \mathbb{R}$ or an internal node $n$ with two children $n.yes$ and $n.no$ decision trees, and a guard $n.g$, which is a Boolean formula involving the input features. Typically, the Boolean formula is a linear inequality of the form $f < v$, where $f$ is a feature and $v$ is a real constant. Given an input $x \in \mathcal{X}$, the (binary) decision tree evaluates it top-down from the root, by evaluating the Boolean formulae in the guards to determine the child to go to, and finally reach a leaf, where it returns the output value of the leaf encountered. For instance, guard $g = f < v$, $g(x)$ is true if $x_f < v$, where $x = (x_f)_{f \in \mathcal{F}}$. This defines the output or outcome $T(x)$ of tree $T$ on input $x \in \mathcal{X}$ as follows: $T(x) = T.val$, if $T$ is a leaf, else, it is recursively evaluated, i.e., if $T.g(x)$ is true, it is $T.yes(x)$ and if $T.g(x)$ is false, it is $T.no(x)$.

**Decision Tree Ensembles**  Rather than relying on a single tree to approximate the underlying function, decision tree ensembles utilize multiple trees, each learning different parts of the problem and then aggregating them to produce a single output. There are many different methods to train decision tree ensembles, and in this paper, we focus on XGBoost (Chen & Guestrin, 2016), a popular gradient-boosting algorithm. Formally, an ensemble model $c = \{T_1, \ldots, T_m\}$ is a set of decision trees. The ensemble model sums up the results of the member trees. We define the outcome of the ensemble model $c$ as $c(x) = \sum_{i=1}^{m} T_i(x)$. We will often refer to this value as the model's raw output. Note that the decision tree ensembles we consider in this paper are always additive (even if we do not explicitly mention it), i.e., they aggregate the trees by summing up the results of individual trees.

**Tree Ensemble Classifiers**  For classification tasks in many real-world applications, the output space $\mathcal{Y}$ is finite, meaning that the set of possible outputs, commonly known as labels in this scenario, is discretized. A tree ensemble classifier $c$ is a decision tree ensemble with an output space $\mathcal{Y} = \{0, 1, \ldots k - 1\}$ where $k$ is the total number of distinct classes. The output of the tree ensemble classifier is acquired by first calculating the raw output of the tree ensemble model $c(x) \in \mathbb{R}$ and then mapping $c(x)$ to its label from $\mathcal{Y}$. We denote the output of the model as $c_{\text{label}} : \mathcal{X} \to \mathcal{Y}$.

A binary tree ensemble classifier is a decision tree ensemble with an output space of size 2. To map the raw output to a binary output, i.e., in $\{0, 1\}$, one way is to set the output to 1 if $c(x) \geq 0$ and 0 otherwise. However, widely used training methodologies (for instance, the XGBoost algorithm introduced in Chen & Guestrin (2016)) require us to calculate an intermediate value by applying

the sigmoid function to the raw output. This can also be interpreted as a measure of the confidence of the model. and is of interest to us. Let $c(x)$ be the raw output of the ensemble for input $x$. To map this to a probability space $[0, 1]$, the sigmoid function is applied on $c(x)$ making the new output: $\sigma(c(x)) = \frac{1}{1+e^{-c(x)}}$. This transformation gives us $\sigma(c(x))$, which represents the probability that the input $x$ belongs to the positive class (1). The closer $\sigma(c(x))$ is to 1, the higher the likelihood that $x$ is a positive example, and vice versa. To calculate the final output ($c_{\text{label}}$), if the output of the sigmoid function is greater than or equal to $0.5$, we assign it to the class (1), and if the output is less than $0.5$, we assign it to the class (0).

## 3 THE SENSITIVITY PROBLEM: MODELING AND HARDNESS

We now define the sensitivity problem for tree ensembles. For an input $x \in \mathcal{X}$ and set of features $F \subseteq \mathcal{F}$, let $x_F$ denote the projection of $x$ onto $F$. When $F = \{f\}$, we use $x_f$ to denote the scalar.

**Definition 3.1.** Given a tree ensemble classifier $c : \mathcal{X} \longrightarrow \mathcal{Y}$, and a set of features $F \subseteq \mathcal{F}$, $c$ is said to be $F$-sensitive, if we can find two inputs $x, x' \in \mathcal{X}$ such that $x_{\mathcal{F} \setminus F} = x'_{\mathcal{F} \setminus F}$ and $c_{\text{label}}(x) \neq c_{\text{label}}(x')$. [1] The sensitivity problem asks whether a given tree ensemble classifier is $F$-sensitive to a given set of features $F$.

One issue with the above definition is that it just requires the two witness inputs to be classified differently but does not take into account their distance from the decision boundary (of the classifier). As explained earlier, classifier learning algorithms, in fact, provide this information. Consider the example in Figure 1, where we have a two-tree $T_1, T_2$ ensemble $c$ that is trained to be a recommendation system for television shows. The input contains three features $f_1, f_2$ and $f_3$ that may take values between 0 and 30 with $\mathcal{Y} = \{0, 1\}$. It is easy to see that $\alpha_1 = (6, 2, 14)$ and $\alpha_2 = (6, 2, 17)$ are two inputs that vary only on feature $f_3$ but have $c(\alpha_1) = 0.1 - 1.2 = -1.1$ while $c(\alpha_2) = 0.7$, i.e., $c_{\text{label}}(\alpha_1) \neq c_{\text{label}}(\alpha_2)$. However, the pair $\beta_1 = (6, 4, 14)$ and $\beta_2 = (8, 4, 14)$ varies only in feature $f_1$ and again has $c_{\text{label}}(\beta_1) \neq c_{\text{label}}(\beta_2)$, but we notice that $c(\beta_1) = 10.1$ and $c(\beta_2) = -9.9$. In a practical scenario, one may argue that $(\beta_1, \beta_2)$ is a much more interesting pair of inputs since it takes an almost sure positive prediction to a quite sure negative prediction, which can potentially be very harmful compared to an unsure prediction getting flipped.

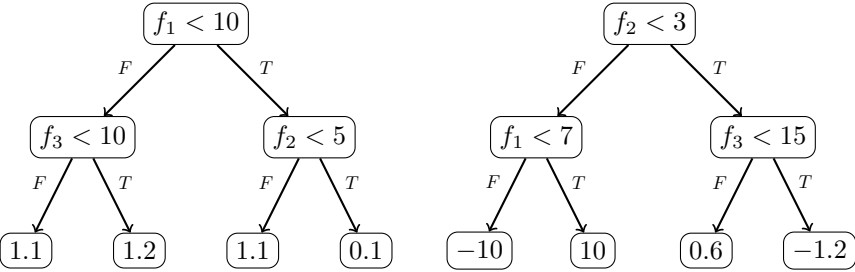

Figure 1: A tree ensemble with two trees $T_1, T_2$ having real-valued (raw) outputs on its leaves.

This motivates a more nuanced definition that is parametrized by the difference using the sigmoid function $\sigma$ that we want to see in the inputs that witness the change in classification with respect to the sensitive features.

**Definition 3.2.** Given a tree ensemble classifier $c : \mathcal{X} \longrightarrow \mathcal{Y}$, a set of sensitive features $F \subseteq \mathcal{F}$ and a parameter $p \geq 0$, $c$ is said to be $(p, F)$-sensitive, if we can find two inputs $x, x' \in \mathcal{X}$ such that $x_{\mathcal{F} \setminus F} = x'_{\mathcal{F} \setminus F}$ and $\sigma(c(x)) \geq 0.5 + p$, $\sigma(c(x')) \leq 0.5 - p$. The $(p, F)$-sensitivity problem asks if a given tree ensemble is $(p, F)$-sensitive with respect to a given set of sensitive features $F$.

It is easy to see that the $F$-sensitivity problem is a special case of the $(p, F)$-sensitivity problem. Hence, for our hardness results, we will focus on the $F$-sensitivity problem. However, for our

---

[1] We note that this problem has been called causal discrimination in Calzavara et al. (2023) and fairness in Dwork et al. (2012). While these are valid applications of the definition, the problem is itself closer to the analysis of sensitive features, and hence, we have called it such. Further, there are many competing definitions of fairness in the ML literature, and while this is certainly one formulation, it is not the only one.

encoding and algorithm in the next section, as well as experimental results later, we will use $(p, F)$-sensitivity.

Sensitivity checking for single tree decision models is known to be in polynomial time (Chen et al., 2019b), but as we show next, this is not the case for general decision tree ensembles. We consider three variants of the problem, based on the size of the set of sensitive features $F$: (i) $|F| = 1$ or the single feature sensitivity problem, (ii) $|F| = k$ for any fixed constant $k$ which we call the $k$-sized subset feature sensitivity problem and (iii) $|F| = |\mathcal{F}|$, the all-feature sensitivity problem. Note that the single feature sensitivity is the same as 1-subset feature sensitivity. Second, the $k$-sized subset feature sensitivity problem is only defined if the number of features in the problem instance i.e., $|\mathcal{F}| \geq k$. Finally, the all-feature sensitivity cannot be seen as a specific instance of the $k$-sized subset sensitivity problem since in the latter, $k$ is a part of the input, while in the former, it is not.

We will now show that all three variants are NP-hard. We start by showing the NP-hardness of the single feature sensitivity problem and obtain the others as corollaries. Before going into our proof, we recall that the robustness verification problem for decision tree ensembles was shown to be NP-hard using a reduction from the so-called evasion problem for decision tree ensembles, which was shown to be NP-hard in Kantchelian et al. (2016) (where, evasion for a given tree ensemble model $c$, asks whether there exists an $x \in \mathcal{X}$ such that $c(x) > 0$). However, while evasiveness can potentially be used to show hardness for the all-feature sensitivity problem, it cannot be easily lifted to show single or fixed subset feature hardness. Instead, we come up with a novel and direct reduction from the 3CNF-SAT problem to show that single feature sensitivity is already NP-hard.

**Theorem 1.** The single feature sensitivity problem, i.e., checking whether a given tree ensemble classifier is $F$-sensitive for $|F| = 1$, is NP-hard.

*Proof.* We will show a reduction from 3CNF-SAT, the classical NP-hard question, which asks, given a Boolean formula in conjunctive normal form (CNF) with 3 variables per clause, whether it is satisfiable. Given an instance $\varphi$ of 3CNF-SAT, let $cl(\varphi)$ be the set of clauses $\{cl_1, cl_2, \ldots, cl_m\}$, with $m = |cl(\varphi)|$ and let $var(\varphi)$ denote the set of variables $\{v_1, v_2, \ldots, v_n\}$, with $n = |var(\varphi)|$. Then from $\varphi$ we start by creating the formula $\varphi' = \varphi \wedge (v_{n+1} \vee v_{n+1} \vee v_{n+1})$ which is also a 3CNF formula with a new variable $v_{n+1}$ and a new clause $cl_{m+1} = (v_{n+1} \vee v_{n+1} \vee v_{n+1})$. Observe that $\varphi$ is satisfiable, i.e., there exists an input $x \in \{0, 1\}^n$ that satisfies $\varphi$ iff $\varphi'$ is satisfiable, i.e., there exists an input $x' \in \{0, 1\}^{n+1}$ that satisfies $\varphi'$. We will now show a reduction to the (single feature) sensitivity problem. That is, we will construct a decision tree ensemble $c$ with depth 3, such that $c$ is 1-feature sensitive iff $\varphi'$ is satisfiable.

In formula $\varphi'$, for every clause $cl_i$, we create a depth-3 decision tree $T_i$ as depicted in Figure 2, where $m + 1 = |cl(\varphi')|$. That is, for each literal (i.e., $v_i$ or $\neg v_i$) in the clause, we add a "true" branch with output $\frac{1}{|cl(\varphi')|}$, and a "false" branch where we either continue to next literal or return $-1$ if there are no more literals left in the clause. This is a slight abuse of notation as our definition earlier had guards of the form $f < v$, but this can easily be adapted. Now, for each literal, if it occurs positively as $v_i$ (resp. negatively as $\neg v_i$), the true (resp. false) branch outputs $\frac{1}{|cl(\varphi')|}$. We form the decision tree ensemble $c$ using the above decision trees with trees enumerated $T_i$ for $i \in \{1, 2, \ldots, m + 1 = |cl(\varphi')|\}$. Note that in this case, the domain of $c$, i.e., $\mathcal{X} = \{0, 1\}^{n+1}$.

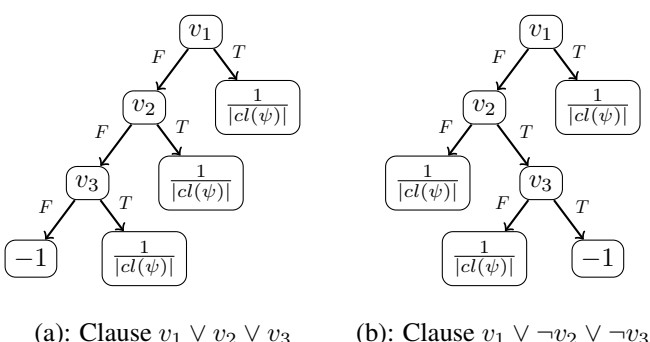

(a): Clause $v_1 \vee v_2 \vee v_3$      (b): Clause $v_1 \vee \neg v_2 \vee \neg v_3$

Figure 2: Given a formula $\psi$ with $|cl(\psi)|$ clauses, each clause is replaced by a tree above.

**Lemma 1.** For all $x \in \{0,1\}^{n+1}$ we have $c_{\text{label}}(x) = 1$ iff $\varphi'(x) = 1$, i.e., $x$ satisfies/models $\varphi'$.

*Proof of Lemma 1.* There are two possible scenarios for an input $x$:

- The input satisfies the 3CNF formula $\varphi'$, i.e., $\varphi'(x) = 1$. In this case, each of the $m + 1$ clauses is satisfied in the input, and thus, for all trees $T_i$ we have $T_i(x) = \frac{1}{m+1}$. Thus, $\sum_{i=1}^{m+1} T_i(x) = 1 > 0 \implies c_{\text{label}}(x) = 1$.

- The input does not satisfy the 3CNF formula $\varphi'$. Thus, a clause exists that is not satisfied by the input. Let that clause be $cl_j$. By the construction of $c$, for the corresponding tree, $T_j(x) = -1$ and for all $i \neq j$, $T_i(x) \leq \frac{1}{m+1}$. Thus, $\sum_{i=1}^{m+1} T_i(x) \leq -1 + \frac{m}{m+1} = \frac{-1}{m+1} < 0 \implies c_{\text{label}}(x) = 0$.

$\blacksquare$

Now, we use the above lemma to prove hardness of sensitivity. More precisely, we will check sensitivity with respect to the single Boolean variable $v_{n+1}$. Call the set of all features $\mathcal{F}$ and the set for sensitivity checking $F = \{v_{n+1}\}$. To complete the proof, we will show that $c$ is $F$-sensitive iff $\varphi'$ is satisfiable.

In one direction, if $c$ is $F$-sensitive, by definition, there exist $x, x' \in \{0,1\}^{n+1}$, with $x_{\perp F} = x'_{\perp F}$ such that $c_{\text{label}}(x) = 1$ and $c_{\text{label}}(x') = 0$. Thus, we immediately infer that there exists $x$ such that $c_{\text{label}}(x) = 1$, which by the above lemma means that $x$ satisfies $\varphi'$ and hence $\varphi'$ is satisfiable. In the other direction, if $c$ is not $F$-sensitive. Then for all $x_{\perp F} \in \{0,1\}^n$, for all possible choices of $x_F, x'_F$, we must have $c_{\text{label}}(x_{\perp F}, x_F) = c_{\text{label}}(x_{\perp F}, x'_F)$. But now, if we consider $x_F = 0$, then the decision tree $T_{m+1}$ will evaluate to $-1$ since $cl_{m+1}[v_{n+1} \mapsto 0] = 0$. As a result, we can conclude that for any $x_{\perp F} \in \{0,1\}^n$, we have $\sum_{i=1}^{m+1} T_i(x_{\perp F}, 0) \leq -1 + \frac{m}{m+1} < 0$ and so $c_{\text{label}}(x_{\perp F}, 0) = 0$. Thus, for any $x_F \in \{0,1\}$, $c_{\text{label}}(x_{\perp F}, x_F) = 0$, which implies that for all $x \in \{0,1\}^{n+1}$, $c_{\text{label}}(x) = 0$. Again, appealing to Lemma 1 above, we can conclude that $\varphi'$ is not satisfiable.

Thus, we have reduced the problem of finding satisfiability of $\varphi'$ to that of checking sensitivity for a feature set of size 1, and hence, the latter problem is NP-hard. $\square$

We can now infer some interesting corollaries. First, we observe that the above proof can be lifted from single feature sensitivity to $k$-sized subset feature sensitivity for any fixed $k$. Given an arbitrary instance of the single feature sensitivity problem, we can construct an instance of a $k$-sized subset feature sensitivity (for $|F| = k$) by introducing $k - 1$ new dummy variables to the problem. These dummy variables are part of our input to sensitivity checking, but they do not affect the tree's output in any way (formally, one way to do this is to have decision tree stumps on these variables that output zero irrespective of the variables' values). Thus, checking for sensitivity for $k$-sized subset of features will be equivalent to checking for sensitivity for just the first feature in the original instance and hence, checking sensitivity with respect to a given fixed size of the subset of features is also NP-hard.

**Corollary 1.** For any fixed constant $k$, the $k$-sized subset feature sensitivity problem for decision tree ensembles is NP-hard.

The above argument holds for any $k$-sized sensitive feature set. But this does not immediately imply that it can be lifted to the case where all features are sensitive, i.e., $|F| = |\mathcal{F}|$. Indeed, one could imagine that the all-feature sensitivity problem could be easier. However, by modifying the proof above, we can show that this problem is also NP-hard. As the proof is very similar, we leave its details to Appendix A.

**Corollary 2.** The all-feature sensitivity problem for decision tree ensembles is NP-hard.

Finally, an important remark is that our NP-hardness proof requires trees in the ensembles that have depth 3. This leaves open the intriguing question of whether the (single/$k$-sized subset/all) sensitivity problem is NP-hard for tree ensembles where the depth of each tree is at most 2. Unfortunately, our

proofs cannot be extended to this case since they rely on the hardness of 3CNF-SAT, which requires 3 variables per clause, which we translate to depth 3 trees. However, 2CNF-SAT is poly-time solvable and hence not useful for showing hardness. We expect that a different technique/reduction/encoding will be needed to resolve this question, and we will leave this for future work. We also note that if we fix the number of trees and vary the depth, or if we fix the depth and vary the number of trees, NP-hardness follows, while if we fix both the number of trees and depth, then the problem is easy.

Above, we considered hardness results. On the other hand, for all the problems mentioned above, we can obtain NP upper bounds. To see this, note that given a candidate solution i.e., values of inputs $x$ and $x'$ that only differ on the set of sensitive features $F$, we can evaluate the decision tree top-down and check whether it is a valid solution or not, i.e., the decision for $x$ differs from the decision for $x'$. Thus, both $F$-sensitivity and $(p, F)$-sensitivity problems are in NP hence they are NP-complete.

## 4 ENCODING THE SENSITIVITY PROBLEM

In this section, we will consider encoding the problem of sensitivity of binary tree ensemble classifier $c$ into solving a set of pseudo-Boolean constraints (Boros & Hammer, 2002), which are arithmetic constraints containing only Boolean variables.

First, we observe that the general $p$-sensitivity problem in Definition 3.2 can be written as the problem of the search of the $p$-sensitive pairs of inputs $x, x'$ for the set of features $F$ as follows

$$\exists x, x', \left( \bigwedge_{f' \in \mathcal{F} \setminus F} x_{f'} = x'_{f'} \right) \wedge \sigma(c(x)) \geq 0.5 + p \wedge \sigma(c(x')) \leq 0.5 - p$$

Since the output of the classifiers is via the sigmoid function, we compute the inverse of the function to compute the required gap between the inputs of the sigmoid function. Let $\delta = \sigma^{-1}(0.5 + p) = \log\left(\frac{0.5+p}{0.5-p}\right)$, where $\sigma^{-1}$ is the inverse of sigmoid. Let $c$ be a decision tree ensemble with $m$ trees $T_1, \ldots, T_m$. The problem translates into

$$\exists x, x', \left( \bigwedge_{f' \in \mathcal{F} \setminus F} x_{f'} = x'_{f'} \right) \wedge \sum_{i=1}^{m} T_i(x) \geq \delta \wedge \sum_{i=1}^{m} T_i(x') \leq -\delta$$

Let us start encoding the above constraints using a pseudo-Boolean encoding.

**Encoding inputs** The range of feature $f$ is $[-\infty, \infty]$. However, $c$ naturally divides the input into segments, which we define as follows. Let $G_f$ be the set of all guards on feature $f$ in $c$. Let the set of thresholds be $C_f = \{v | f < v \in G_f\}$ sorted in ascending order and let $k_f = |C_f|$. The range of $f$ is divided in $k_f + 1$ segments by $C_f$. We can encode the segments using $k_f$ bits. For $0 \leq j < |C_f|$, let bit $b_{1fj}$ indicate that feature $f$ is less than $C_f[j]$ in input $x$ and let bit $b_{2fj}$ indicate that feature $f$ is less than $C_f[j]$ in input $x'$. We need to include constraints which encode that the boundaries are in increasing order, as follows

$$b_{qfj} \Rightarrow b_{qf(j+1)} \tag{1}$$

We also need to say for each $f \notin F$, $x$ and $x'$ will agree. Therefore, for each $j \in \{1 \ldots k_f\}$, we add

$$b_{1fj} = b_{2fj} \tag{2}$$

**Encoding tree** We need to encode the structure of the trees in constraints. Let $t_{qin}$ indicate that the node $n \in T_i$ is visited when evaluating $T_i(x_q)$. Let the root node of $T_i$ be denoted by $r_i$. Since the roots of all the trees are necessarily visited, the following bits are always true

$$t_{qir_i} \tag{3}$$

For each internal node $n \in T_i$, let $(f < v) = n.g$ such that $C_f[j] = v$ for some $j$. We need to say that if $n$ is visited, and $(f < v)$ is true, then $n.yes$ is visited otherwise, $n.no$ is visited.

$$(t_{qin} \wedge b_{qfj} \Rightarrow t_{qi(n.yes)}) \wedge (t_{qin} \wedge \neg b_{qfj} \Rightarrow t_{qi(n.no)}) \tag{4}$$

For each tree $T_i$, we may also *optionally* add the following constraints to help the solver to know that, at most one leaf of $T_i$ can be visited by the input. We denote the set of leaf nodes for a tree $T_i$ by $T_i.leaves$

$$\sum_{n \in T_i.leaves} t_{qin} = 1 \tag{5}$$

**Encoding output** We now need to encode the condition that the sum of the tree outputs is greater than $\delta$ for $x$ and less than $-\delta$ for $x'$. Remember that the label of a leaf $n \in T_i$ is a real value with infinite precision. We discretize this real value into steps of size $1/\alpha$, where $\alpha$ is the precision factor in our encoding. For the encoding of $x$, we multiply the leaf value $n.val$ by $\alpha$ and take the ceiling of the result. This constant integer is then multiplied by the corresponding $t_{1in}$. The ceiling operation increases the sum, which is then compared to the bounds derived from the floor of $\alpha\delta$. Similarly, we impose constraints for the output of $x'$.

$$\left( \sum_{i=1}^{m} \sum_{n \in T_i.leaves} t_{1in} \lceil \alpha t.val \rceil \right) \geq \lfloor \alpha\delta \rfloor \wedge \left( \sum_{i=1}^{m} \sum_{n \in T_i.leaves} t_{2in} \lfloor \alpha t.val \rfloor \right) \leq \lceil -\alpha\delta \rceil \tag{6}$$

The above equations are the set of pseudo-Boolean constraints.

Our encoding of the problem differs significantly from the one presented in Ignatiev et al. (2020a), which is a direct SMT encoding of trees and their verification query. The problem we address is similar to the knapsack problem, and an appropriate encoding for it seems to be through pseudo-Boolean constraints. Encoding the problem as SMT constraints, as done by Ignatiev et al. (2020a), may result in a loss of structural information, preventing the solver from fully leveraging the pseudo-Boolean nature of the constraints. We solve these constraints using a pseudo-Boolean solver to check for $p$-sensitivity. We have the the following guarantees for the result that we obtain.

**Theorem 2** (Completeness and $\alpha$-soundness).      1. If conjunction $(1) \wedge (2) \wedge (3) \wedge (4) \wedge (5) \wedge (6)$ is unsatisfiable then the classifier $c$ is not $(p, F)$-sensitive.

2. Otherwise, there is a counterexample pair $(x, x')$ such that $c(x) \geq \delta - \frac{m+1}{\alpha}$ and $c(x') \leq -\delta + \frac{m+1}{\alpha}$, where $\delta = \log\left(\frac{0.5+p}{0.5-p}\right)$ and $\alpha$ is the precision parameter of our analysis.

*Proof sketch.* Since the floor and ceiling operations are in conservative directions, 1 is true. In case of satisfiable constraints, there will be exactly $m$ bits that are 1 in the sum in (6). The pre-sigmoid output in the reported counterexamples will be closer to zero by amount $(m+1)/\alpha$ in $\delta$, due to $m$ ceiling/floor operations on the left-hand side of the inequalities and one ceiling/floor operation on the right-hand side in (6). Therefore, 2 holds. $\qquad\square$

As we increase $\alpha$, the precision of our counterexamples improves. Thus, we can prove the following corollary, which states that if $\alpha$ is sufficiently large, the counterexamples will be precise.

**Corollary 3.** There exists an $\alpha$ such that every counterexample pair $(x, x')$ satisfying the above pseudo-Boolean formula will satisfy $\sigma(c(x)) \geq 0.5 + p$ and $\sigma(c(x')) \leq 0.5 - p$.

*Proof Sketch.* In the non-approximated constraint $\sum_{i=1}^{m} \sum_{n \in T_i.leaves} t_{1in} t.val > \delta$, consider the difference between $\delta$ and the sum closest to, but less than $\delta$. If this difference is larger than $(m+1)/\alpha$, then there can be no extra counterexamples due to the imprecision of the encoding by Theorem 2. $\qquad\square$

## 5   EXPERIMENTS

In this section, we present our tool, SENSPB[2], which implements the above method for $p$-sensitivity checking. The tool is developed in Python and utilizes Z3 (de Moura & Bjørner, 2008) as its backend pseudo-Boolean solver. We also tried a dedicated pseudo-Boolean solver (Elffers & Nordström, 2018), but its performance was similar, so we kept to Z3. Our tool accepts as input a binary classifier

---

[2]https://github.com/Arhaan/SensPB

| Benchmark Name | Details | | | SENSPB time taken (s) | | | SMT |
|---|---|---|---|---|---|---|---|
| | #Trees | Depth | #Feat | Min | Max | Average | |
| Breast cancer robust | 4 | 5 | 11 | 2.72 | 2.81 | 2.76 | 2.76 |
| Breast cancer unrobust | 4 | 6 | 11 | 2.75 | 2.82 | 2.8 | 2.8 |
| Diabetes robust | 20 | 5 | 9 | 3 | 3.2 | 3.02 | 3.443 |
| Diabetes unrobust | 20 | 5 | 9 | 3.4 | 23 | 5.8 | 57.2 |
| Cod-rna unrobust | 80 | 4 | 8 | 7.2 | 12.9 | 8.3 | 106 |
| Binary MNIST robust | 50 | 6 | 784 | 14.6 | 15.3 | 14.9 | TO |
| Higgs unrobust | 100 | 8 | 28 | 130 | TO | 1188 | TO |
| IJCNN robust | 60 | 8 | 23 | 16 | TO | 330.7 | TO |
| Synthetic 1 | 100 | 6 | 10 | 5.36 | 5.85 | 5.57 | TO |
| Synthetic 2 | 125 | 6 | 10 | 6 | 6.35 | 6.2 | TO |
| Synthetic 3 | 150 | 6 | 10 | 7.09 | 8.19 | 7.36 | TO |
| Synthetic 4 | 175 | 6 | 10 | 6.25 | 6.57 | 6.37 | TO |
| Synthetic 5 | 200 | 6 | 10 | 4.40 | 124.51 | 16.48 | TO |

Table 1: Times taken for verifying or countering sensitivity of all singular feature sets. The Min, Max and Averages in SENSPB times are taken by running the tool with different features of the benchmark tree ensembles as the sensitive feature. More information on these experiments is available in Appendix B.

XGBoost model, a set of features for which sensitivity is being assessed, the parameter $p$, and a precision parameter $\alpha$. If the model is not sensitive, the tool outputs "pass". Otherwise, it returns a pair of inputs that demonstrate $p$-sensitivity on the specified features.

To assess our method, we begin by running our tool on a set of XGBoost models from Chen et al. (2019b). Additionally, to evaluate the performance of our tool, we train XGBoost models with varying numbers of ensemble trees on 100,000 randomly generated data samples. We did not run experiments for (additive) Random Forests separately since, from the point of view of our encoding, they are equivalent to GBDT models. We ran the experiments on an Ubuntu machine with 20 1.3GHz cores, which has 64GB RAM.

There have been several tools (Devos et al., 2021; Törnblom & Nadjm-Tehrani, 2020; Chen et al., 2019b; Ignatiev et al., 2020a; Calzavara et al., 2023; Kantchelian et al., 2016) that implement different variants of verification for tree ensembles. In our experiment, we compare SENSPB with the closest approach in VERITAS (Devos et al., 2021) and an SMT-based approach presented in Ignatiev et al. (2020a). We used our own implementation of the SMT-based approach with Z3 (de Moura & Bjørner, 2008) as the SMT solver. We did not compare with Calzavara et al. (2023) as it only supports random forest trees and we were unable to make it work with XGBoost models. Finally, we did not include a comparison with Kantchelian et al. (2016) since VERITAS has already demonstrated superior performance over this tool, albeit for the problem of local robustness verification.

We ran SENSPB on the benchmarks, and the results are presented in Table 1. For each benchmark ensemble, in one experiment we pick a feature $f$ and run SENSPB on the ensemble to check whether the classifier is $p$-sensitive to $f$. We repeat this experiment over all possible $f$, and the maximum, minimum and average time taken by us for termination is reported. We set a timeout of 1 hour for each experiment. In our experiments, we have set gap $p = 0.15$ and precision $\alpha = 10 \times |\#\text{Trees}|$. For the SMT solver-based approach, our experimental setup is the same as SENSPB and we report the average time taken. More experiments can be found in Appendix C.

VERITAS does not solve the sensitivity problem directly. We instead ask VERITAS to maximise the difference between the outputs produced by two inputs which differ only in a feature. We define the bounds found by VERITAS being "better than" the ones found by SENSPB if the difference in the upper and lower bounds found by VERITAS is greater than $2p$. Note that this is a very relaxed definition since the bounds found by VERITAS might have a larger spread but might still be lying in the same output class. Since we can stop VERITAS anytime and observe the best solution found till then, we considered two kinds of experiments for fine-grained comparisons between the performance of SENSPB and VERITAS.

| Benchmark Name | Time Taken (in seconds) | | Accuracy comparison | | | |
|---|---|---|---|---|---|---|
| | **SENSPB** | **VERITAS** | **1x** | **2x** | **5x** | **3600** |
| Breast cancer robust | 2.7 | 2.5 | 81.82% | 81.82% | 81.82% | 81.82% |
| Breast cancer unrobust | 2.7 | 2.53 | 45.45 % | 45.45% | 45.45% | 45.45% |
| Diabetes robust | 3.0 | 3.2 | 77.78% | 77.78% | 77.78% | 77.78% |
| Diabetes unrobust | 5.9 | 198.1 | 55.56% | 66.67% | 77.78% | 88.89% |
| Cod-rna unrobust | 8.3 | 346.2 | 0.00% | 0% | 25.00% | 37.50% |
| Binary-mnist robust | 14.9 | TO | TO | TO | TO | TO |
| Higgs unrobust | 1188 | TO | TO | TO | TO | TO |
| IJCNN unrobust | 330 | OOM | TO | TO | 0% | OOM |

Table 2: 1) Runtime Comparison by letting VERITAS run on benchmarks with a timeout of 3600s. 2) Accuracy analysis of VERITAS with the timeouts set to different values, depending on the time it took SENSPB. The percentages recorded represent that fraction of features where VERITAS gave better than or equivalent results as compared to SENSPB. For Time Analysis, TO implies that VERITAS timed out without reaching an optimal difference, while in the Accuracy measurements, TO means that VERITAS timed out without producing a single valid solution.

Firstly, we asked if we fix a timeout of 3600s, how does the performance of VERITAS compare with SENSPB, i.e., how long does it take VERITAS to reach an optimal solution? The results are present in the Time Analysis part of Table 2. For the Binary MNIST robust and the Higgs unrobust benchmark ensembles, VERITAS always times out without producing a single solution. For IJCNN robust, VERITAS runs out of memory after roughly 900s.

Secondly, we asked if we ran VERITAS for the time relative to the time taken by SENSPB, what would be the relative performance? Let the time taken by SENSPB be $x$. We run VERITAS for sensitivity analysis, one feature at a time, with the following timeouts: $x$, $2x$, $5x$ and $3600$. We look at the bound produced by VERITAS at the end of the timeout and compare this bound to the bound found by SENSPB. In the accuracy comparison section of Table 2, we report the percentage of features in which VERITAS performed better than or equivalent to SENSPB. As expected, on increasing the time given to VERITAS, it starts performing better on more and more features, e.g., Diabetes unrobust and Cod-rna unrobust. However, even on running VERITAS for 3600s, there are many features in which SENSPB performs better. As noted earlier, for our three largest benchmarks, VERITAS does not produce a single solution in the time given.

Our experiments clearly demonstrate that our pseudo-Boolean encoding significantly outperforms both the standard encoding and output configuration-based approaches by an order of magnitude. Given that the problem is NP-hard, SMT-based approaches are likely to surpass output configuration-based methods unless those methods are highly optimized. An SMT solver typically uses CDCL along with Simplex to solve the problem but may not fully exploit the specialized nature of our problem, particularly the limited role of arithmetic during bit summation at the output. As a result, pseudo-Boolean solvers specifically designed for such problems are expected to deliver the best performance. Our encoding allows us to not only use pseudo-Boolean solvers but also provide new sets of benchmarks for the solvers. The availability of the benchmarks would likely improve the performances of the solvers.

## 6 CONCLUSION

In this paper, we investigated the sensitivity problem in two variants, the exact and a parametrized version. We presented new hardness proofs as well as an efficient encoding into pseudo-Boolean constraints. Our implementation allowed us to exploit the recent advances in pseudo-Boolean solvers to solve the $p$-sensitivity problem. We successfully addressed XGBoost models of practical sizes, including scores of features, hundreds of trees, and a depth of 6-8. We believe that our work, especially the pseudo-Boolean encoding, opens a new direction for scalable solutions for sensitivity and general verification problems for tree ensembles. For instance, an immediate extension would be to consider other tree ensemble models, such as random forests. Another natural future direction is also towards the question of multiclass labels, e.g., as done in Devos et al. (2024) for robustness verification.

ACKNOWLEDGEMENTS

We acknowledge the SBI Foundation Hub for Data Science & Analytics, IIT Bombay for supporting the work done in this project.

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

# A  ADDITIONAL THEORETICAL RESULTS

## A.1  PROOF FOR COROLLARY 2

The proof of Theorem 1 can be directly lifted with some minor changes to prove Corollary 2. Instead of checking sensitivity for $F = \{v_{n+1}\}$, we check sensitivity for $F = \mathcal{F}$ in the same setting. The first direction holds with the same argument as before. The reverse direction also holds with the same argument with the following changes.

- $x_{\perp F}$ is empty

- $x_F \in \{0, 1\}^{n+1}$ instead of $\{0, 1\}$

However, for the sake of completeness, we give the full proof below.

*Proof.* As before, we show a reduction from 3CNF-SAT. Given an instance $\varphi$ of 3CNF-SAT, let $cl(\varphi)$ be the set of clauses $\{cl_1, cl_2, \ldots, cl_m\}$, with $m = |cl(\varphi)|$ and let $var(\varphi)$ denote the set of variables $\{v_1, v_2, \ldots, v_n\}$, with $n = |var(\varphi)|$. Then from $\varphi$ we start by creating the formula $\varphi' = \varphi \land (v_{n+1} \lor v_{n+1} \lor v_{n+1})$ which is also a 3CNF formula with a new variable $v_{n+1}$ and a new clause $cl_{m+1} = (v_{n+1} \lor v_{n+1} \lor v_{n+1})$. Observe that $\varphi$ is satisfiable, i.e., there exists an input $x \in \{0, 1\}^n$ that satisfies $\varphi$ iff $\varphi'$ is satisfiable, i.e., there exists an input $x' \in \{0, 1\}^{n+1}$ that satisfies $\varphi'$. We will now show a reduction to the (single feature) sensitivity problem. That is, we will construct a decision tree ensemble $c$ with depth 3, such that $c$ is 1-feature sensitive iff $\varphi'$ is satisfiable.

In formula $\varphi'$, for every clause $cl_i$, we create a depth-3 decision tree $T_i$ as depicted in Figure 2, where $m + 1 = |cl(\varphi')|$. That is, for each literal (i.e., $v_i$ or $\neg v_i$) in the clause, we add a "true" branch with output $\frac{1}{|cl(\varphi')|}$, and a "false" branch where we either continue to next literal or return $-1$ if there are no more literals left in the clause. For each literal, if it occurs positively as $v_i$ (resp. negatively as $\neg v_i$), the true (resp. false) branch outputs $\frac{1}{|cl(\varphi')|}$. We form the decision tree ensemble $c$ using the above decision trees with trees enumerated $T_i$ for $i \in \{1, 2, \ldots, m + 1 = |cl(\varphi')|\}$. Note that in this case, the domain of $c$, i.e., $\mathcal{X} = \{0, 1\}^{n+1}$. Remember that Lemma 1 says that for all $x \in \{0, 1\}^{n+1}$ we have $c(x) = 1$ iff $\varphi'(x) = 1$, i.e., $x$ satisfies/models $\varphi'$.

Now, we use Lemma 1 to prove the hardness of sensitivity. More precisely, we will check sensitivity with respect to the singleton Boolean variable $v_{n+1}$. Call the set of all features $\mathcal{F}$ and the set for sensitivity checking $F = \mathcal{F}$. To complete the proof, we will show that $c$ is $F$-sensitive iff $\varphi'$ is satisfiable.

In one direction, if $c$ is $F$-sensitive, by definition, there exist $x, x' \in \{0, 1\}^{n+1}$, such that $c(x) = 1$ and $c(x') = 0$. Thus, we immediately infer that there exists $x$ such that $c(x) = 1$, which by Lemma 1 means that $x$ satisfies $\varphi'$ and hence $\varphi'$ is satisfiable. In the other direction, if $c$ is not $F$-sensitive. Then for all possible choices of $x_F, x'_F$, we must have $c(x_F) = c(x'_F)$. But now, if we consider $x_{v_{n+1}} = 0$, then the decision tree $T_{m+1}$ will evaluate to $-1$ since $cl_{m+1}[v_{n+1} \mapsto 0] = 0$. As a result, we can conclude that for any $x_{F \setminus \{v_{n+1}\}} \in \{0, 1\}^n$, we have $\sum_1^{m+1} T_i(x_{F \setminus \{v_{n+1}\}}, 0) \le -1 + \frac{m}{m+1} < 0$ and so $c(x_{F \setminus \{v_{n+1}\}}, 0) = 0$. Thus, for any $x_F \in \{0, 1\}^{n+1}$, $c(x_F) = 0$, which implies that for all $x \in \{0, 1\}^{n+1}$, $c(x) = 0$. Again, appealing to Lemma 1, we can conclude that $\varphi'$ is not satisfiable.

Thus, we have reduced finding satisfiability of $\varphi'$ to checking sensitivity for the whole input feature set, and hence, the latter problem is NP-hard.

$\square$

An interesting question that arises from the above proof is the requirement of the new clause $cl_{m+1}$. What we require is an input which does not satisfy $\varphi'$. If there is no such input, then even when the decision tree ensemble is insensitive to the set $F$, the 3CNF formula $\varphi'$ can be satisfiable. Thus, to ensure such an input exists, we add the clause $cl_{m+1}$.

## A.2 HARDNESS OF THE DIFFERENTIATING INPUT PROBLEM

From the same construction of trees in Theorem 1, we can show that a novel yet interesting problem, which we call the differentiating input problem, is also NP-hard.

**Definition A.1.** Given two tree ensemble classifiers $c : \mathcal{X} \longrightarrow \mathcal{Y}$ and $c' : \mathcal{X} \longrightarrow \mathcal{Y}$, we say $x \in \mathcal{X}$ is a differentiating input for them if $c(x) \neq c'(x)$. Given two tree ensemble classifiers $c, c' : \mathcal{X} \longrightarrow \mathcal{Y}$, the differentiating input problem asks if there exists a differentiating input, i.e., $\exists x \in \mathcal{X}, c(x) \neq c'(x)$.

**Corollary 4.** The differentiating input problem for decision tree ensembles is NP-Hard.

*Proof.* We will show a reduction from 3CNF-SAT to this problem. Given an arbitrary 3CNF-SAT problem, we can create a decision tree classifier that solves this problem. Given an instance $\varphi$ of 3CNF-SAT, let $cl(\varphi)$ be the set of clauses $\{cl_1, cl_2, \ldots, cl_m\}$, with $m = |cl(\varphi)|$. Then, for each clause $cl_i = l_1 \vee l_2 \vee l_3$ in the 3CNF-SAT problem, create a decision tree, $T_i$ such that $T_i(x) = \frac{1}{m}$ if $x$ satisfies the clause and $T_i(x) = -1$ otherwise. Two examples for the same are shown in Figure 2(a) and 2(b). The ensemble $c = \{T_1, T_2, ..., T_m\}$ outputs a positive class (+1) if and only if the 3CNF-SAT formula was satisfiable as shown in the proof of Theorem 1.

Finally, we create another tree ensemble $c'$, which always returns a negative class(-1). Thus, asking whether there is a differentiating input for these tree ensembles is equivalent to asking whether the 3CNF-SAT formula was satisfiable, thus completing the hardness proof.

## B MORE INFORMATION ON THE EXPERIMENTS

The following table gives the fraction of all the features to which the benchmark trees are singularly sensitive.

| Benchmark Name | Number of Features Ran On | Percentage of Sensitive Features |
|---|---|---|
| Breast cancer robust | 11 | 18.2 |
| Breast cancer unrobust | 11 | 36.4 |
| Diabetes robust | 9 | 44.4 |
| Diabetes unrobust | 9 | 80 |
| Cod-rna unrobust | 8 | 100 |
| Binary mnist robust | 10 | 0 |
| Higgs unrobust | 10 | 100 |
| IJCNN robust | 23 | 100 |

Table 3: Percentage of Sensitive Features

## C ADDITIONAL EXPERIMENTS

We run experiments to understand the effects of changing $p$ and $\alpha$ (separately) on the running time of our algorithm. For these, we work only on the IJCNN robust benchmark, which has 60 trees, with a maximum depth of 80 and 23 features and is a good representative of the kind of ensembles we aim to verify. While changing $p$, we keep the value of $\alpha$ fixed to the value we used in the main experiments (i.e. $10 \times |\#\text{Trees}|$). Likewise, while varying $\alpha$, we keep $p$ fixed to $0.15$. These results are present in Tables 4 and 5. We also give a table detailing the fraction of features where SENSPB performs better than VERITAS, see Table 6.

| p | Time (s) |
|------|----------|
| 0.1 | 15.8 |
| 0.15 | 16 |
| 0.2 | 17.0 |
| 0.4 | 1100.0 |
| 0.45 | 923.4 |

Table 4: Effect of changing gap

| $\alpha$ | Time (s) |
|----------|----------|
| 100 | 16.26 |
| 200 | 15.79 |
| 500 | 15.79 |
| 700 | 16.17 |
| 1000 | 15.67 |
| 1500 | 22.39 |
| 2000 | 19.01 |
| 5000 | 19.63 |
| 100000 | 16.58 |
| 1000000 | 16.71 |

Table 5: Effect of changing precision

| Benchmark Name | VERITAS better | %SENSPB better |
|----------------|----------------|----------------|
| Breast cancer robust | 81.82% | 18.18% |
| Breast cancer unrobust | 45.45 % | 54.55% |
| Diabetes robust | 77.78% | 22.2% |
| Diabetes unrobust | 55.56% | 44.44% |
| Cod-rna unrobust | 0.00% | 100% |
| Binary-mnist robust | TO | 100% |
| Higgs unrobust | TO | 100% |
| IJCNN unrobust | TO | 100% |

Table 6: Comparison between the features where VERITAS finds a better bound than SENSPB. We first run SENSPB and then run VERITAS for the same amount of time. We look at the best results produced by VERITAS in this time limit and use that value for the comparison.

