# OpenReview forum: "Sensitivity Verification for Additive Decision Tree Ensembles"
_ICLR.cc/2025/Conference — ICLR 2025 Poster_

### Official Review · Reviewer_XnQM · 2024-10-29

**Soundness:** 3
**Presentation:** 2
**Contribution:** 2
**Rating:** 6
**Confidence:** 4

**Summary:**

In the context of interpretable machine learning, this paper investigates the problem of "feature sensitivity" in tree ensembles. Specifically, given a classifier $ c $ and a set of features $ F $, the goal is to determine whether there exists an instance $ x $ such that the classification of $x$ by $c$ changes when only the values of the features in $ F $ are modified. The paper establishes that this problem is NP-complete for gradient-boosted trees, even when $ F $ consists of a single feature. The authors also explore an extension of this problem that considers the confidence associated with class changes, which is also shown to be NP-complete. To deal with this source of complexity, the authors propose a Pseudo-Boolean encoding for the feature sensitivity task, which they empirically validate on several benchmarks.

**Strengths:**

**Novelty:**
Arguably, the main strength of this paper lies the novelty of the complexity results for deciding feature sensitivity when the predictive model is a binary-class gradient boosted tree. Notably, the fact that the problem remains NP-hard even in the case where the sensitivity analysis is reduced to a single feature is intriguing.

**Weaknesses:**

**Clarity:**
The scope of the paper is quite challenging to define. As indicated in the title, summary, and introduction, it seems that the authors aim to examine feature sensitivity for “tree ensembles,” which is a broad category that includes several model classes, such as gradient-boosted trees (GBTs) and random forests (RFs). This is further illustrated in Section 2, where the authors attempt to provide a general definition of tree ensembles (Lines 130-136). However, upon reviewing the technical sections of the study (Sections 3-5), it becomes clear that the results are exclusively related to GBTs. This focus is reiterated toward the end of the introduction (starting at Line 65), where the authors clarify their exclusive emphasis on GBTs.

Therefore, it is essential for the authors to establish a clear consensus on the scope of the paper in the revised version. If they intend to concentrate solely on GBTs, this specificity should be highlighted throughout the paper, beginning with the title and summary. The definition of tree ensembles in Section 2 should also be replaced with a definition of GBTs. Conversely, if the authors believe that their theoretical results could be easily extended to other tree ensemble classes (e.g., random forests), then the proofs in Section 4 should be modified accordingly (see Question 2).

At present, the definition of tree ensembles in Section 2 is somewhat unclear. If we denote by $c(x)$ the sum of the decisions $T_i(x)$ made by each of the $m$ decision trees $T_i$, then what class in $\mathcal{F}$ is predicted by $c$ on $x$? This is evident for GBTs when $\mathcal{F} = \\{-1,+1\\}$, using the sigmoid function $\sigma$, but it remains ambiguous for other classes of tree ensembles. If the authors’ goal is to address various classes of ensemble models, it is important to provide a comprehensive definition of “tree ensemble” that can be instantiated for both GBTs and random forests.

**Significance:**
The main theoretical contribution of this paper lies in Theorem 2, which establishes that single feature sensitivity is NP-hard for GBTs. Other NP-hardness results presented in this study stem from this case. While this finding is commendable, I remain unsure if it is sufficient for ICRL. To enhance the theoretical significance of this study, it would be beneficial to investigate whether this problem and its variants are W[1]-hard or if some of them are fixed-parameter tractable (FPT), with the parameter of interest being the maximum depth of the trees (See Question 1). Establishing hardness results for W[1] would indeed bolster the justification for the constraint programming approach discussed in Section 4.

**Questions:**

(1) Can the NP-hardness of single feature sensitivity (Theorem 2) or subset feature sensitivity (Theorem 1) for GBTs be extended to W[1]-hardness?

(2) Alternatively, can these hardness results for GBTs be extended to random forests?

---

> ### Author Response · Authors · 2024-11-18
> **Regarding Scope of the Paper, parametric complexity and extension to random forests.**
>
> Thanks for the careful reading, in-depth comments and interesting questions. Below we answer the specific questions asked.
>
> A1: Studying the W[1]-hardness of the problems involved does seem to be a very interesting extension to our work. We will need to think more about that. At the same time, it is interesting to note that there exist constant time algorithms for ensembles when both the maximum depth and the number of trees is bounded. This is because, for such instances, the number of features is necessarily bounded (since the number of internal nodes is bounded). We can check for sensitivity by brute-forcing all possible consistent assignments for the internal nodes till we find a sensitive example. From this simple observation, we can already infer some parametrized results. For instance, we can conclude that if the depth of the tree is bounded, then the problem becomes FPT in the number of trees. Similarly, if the number of trees is bounded, then the problem becomes FPT in the depth of the tree.
> Going beyond this is certainly interesting but perhaps beyond the scope of this paper. In this paper our focus was to (a) establish the problem and its hardness, (b) develop an effective encoding and (c) show the practicality of the encoding via experiments. As detailed in our response to Reviewer [1wTC], the encoding via pseudo-Boolean constraints is not just novel, it is the reason we are able to make use of the advances in solver-technology and obtain results outperforming the state of the art.
>
> A2: Regarding whether we can lift the hardness results to random forests, the answer is Yes, the hardness results can indeed be directly extended to (additive) random forests. We have detailed this answer as a general response as multiple reviewers were interested in it. We hope this also answers the question about the scope of the paper and why we chose to present it as we did.

---

> > ### Comment · Reviewer_XnQM · 2024-11-20
> >
> > I appreciate the authors' response. While setting limits on both the depth of the trees and the number of trees in the ensemble is a good starting point, it is not realistic in practice. GBT models typically consist of hundreds or even thousands of small decision trees. Therefore, the key question here is: given a small depth $ k $ and a large number of trees $ m $, does the single-feature sensitivity problem for GBTs allow for a $ 2^k m^{O(1)} $ time algorithm? If the answer is "No," by proving that the problem is W[1]-hard then, arguably, the best we can hope for is to solve it—albeit without polynomial-time guarantees—using an efficient constraint encoding, such as the PB encoding proposed in the paper.

---

> > > ### Author Response · Authors · 2024-11-27
> > >
> > > Thanks for the question! In fact, our proof already shows that "for a (small) depth $k$ and a (large) number of trees $m$, the single-feature sensitivity problem for GBTs cannot be solved in time $2^{f(k)}\cdot m^{O(1)}$ (for any function $f$ that depends only on $k$), unless P=NP”.
> > > To see this, note that our main Theorem (Theorem 2 in the paper) shows that, “Given a tree ensemble of GBTs all of whose trees are of finite depth (in the current version of the paper this depth is 4, while the revised/improved proof above improves it to 3), the single feature sensitivity problem is NP-hard. Now, suppose there exists an algorithm for single feature sensitivity that ran in time $2^{f(k)}\cdot m^{O(1)}$, where $k$ is the max depth of trees, then for the finite depth case (regardless of the depth being 3 or 4), this would result in a polytime algorithm, which would imply P=NP. Thus, we can’t realistically hope for anything better than using efficient constraint-based encodings, such as our PB algorithm.
> > > (Of course, one can still ask if there are direct reductions from W[1]-hard problems, such as k-dominating set or k-clique, and while this is certainly interesting, it would require completely new reductions and we leave this for future work.)

---

### Official Review · Reviewer_Mrks · 2024-11-02

**Soundness:** 3
**Presentation:** 3
**Contribution:** 3
**Rating:** 6
**Confidence:** 4

**Summary:**

The paper addresses the sensitivity verification problem of GBDTs by formalizing a pseudo-Boolean encoding. The authors claim their approach can achieve significant performance gains over existing sensitivity verification tools, specifically for GBDTs. However, the paper lacks a comprehensive presentation, robust mathematical formulation, and systematic experiments. As such, it would need major revisions to be accepted.

# Discussion phase
After a deep discussion phase during which the authors addressed all my weaknesses and improved their work, including a better result, I increased my overall score to 6/10. Specifically, I've changed only the scores while leaving the original review; I could also change the review if that's mandatory.

**Strengths:**

- **S1:** The paper addresses an important issue aimed at verifying the sensitive features of GBDTs.
- **S2:** Mathematical proofs, which I would have expected to have a better presentation, but I appreciate the effort.
- **S3:** The authors present an approach using pseudo-Boolean encoding, arguing that it is more suitable than other methods.

**Weaknesses:**

- **W1:** The notation is inconsistent and confuses readers, especially those who are unfamiliar with the topic. For example, input examples are denoted by $x$ and $x'$ in Def. 3.1, but then they are $x_1$ and $x_2$ in Def. 3.2; furthermore, in the proof of Th. 2 instances are symbolized $a$ and $b$ (a very unhappy notation). Similarly, trees are defined as $T$ in Section 2 but then symbolized with $D$ in the proof of Th. 1.
- **W2:** Certain mathematical expressions and definitions are unclear or incorrect. For example, $F = f$ instead of $F = \\{f\\}$ detracts from the overall comprehensibility. As another example, the set minus ($\setminus$) operation is properly used in Def. 3.1 but then in Section 4 is not used; that is, the authors use minus ($-$). A further example is when the authors use $\mathcal F = F \bigcup f$, which, again, is not a proper mathematical definition; it should perhaps be $\mathcal F = F \cup \\{ f \\}$.
- **W3:** The proofs suffer from inconsistent symbols, poor spacing, and cluttered notation, making the logical progression of arguments hard to follow. For example, while reading the proof of Th. 1, I had to stop and skip such proof because the notations became heavy, and my overall cognitive overload intensified. To give a precise idea of how it felt: in $c'(x) = -n$, $n$ is the same $n$ as the number of decision trees in the ensemble (line 205)? What is the $\Longrightarrow$ in the notation? What is $\wedge$? Etc. As you can see, there are too many doubts that the reader has at this point. I assumed the proof was correct, even though I hadn't thoroughly reviewed it. The problem with this way of writing the proof is that the authors mixed textual explanations with mathematical ones in an unpleased way. This point is very critical.
- **W4:** The experimental results lack clarity; terms like "maximum, minimum, and average" time are used ambiguously without explaining what they represent or if multiple runs were performed.
- **W5:** Despite the general title, the paper focuses solely on GBDTs, excluding other ensemble methods like random forests without justification. If that's not the case, I expected to see experiments with random forests as well.
- **W6:** The paper contains grammatical errors, inconsistent abbreviations (e.g., "wrt" vs. "w.r.t."), and informal language, which detracts from the overall presentation.

**Questions:**

- **Q1:** Could you ensure consistent notation throughout, especially in proofs? E.g., $x$ and $x'$ vs $x_1$ and $x_2$.
- **Q2:** Why did you limit the work to GBDTs? Would your framework extend to other ensemble methods, such as random forests?
- **Q3:** Could you clarify the connection between sensitivity and $(p)$-sensitivity, especially after Definition 3.2? The special-case/general-case distinction you imply needs a clearer mathematical basis, and it is *not* easy to see (i.e., one uses the sigmoid function $\sigma$ and the other does not).
- **Q4:** Why do you use the sigmoid function $\sigma$ in Def. 3.2, and then decide to drop such "complexity" in Section 4 when you present the encoding? I would either drop it from the definition or introduce it in the encoding; either way would be acceptable, but please avoid the asymmetric treatment.
- **Q5:** What does "maximum, minimum, and average time" mean in your experiments? Were these metrics based on repeated runs, and if so, could you clarify the settings?

---

> ### Author Response · Authors · 2024-11-18
> **Response to Questions**
>
> Thanks for the review and critical remarks. We understand that there were some difficulties in parsing the notation/presentation, and we will improve it and make the desired changes, including a careful grammatical error check. Below, we address all the questions asked:
>
> A1. Regarding making notations consistent: We will improve the presentation to fix the identified issues.  In particular we will use $x,x’$ instead of $x_1,x_2$ throughout. As pointed out by another reviewer, this has the added advantage of avoiding some “double-subscripts”. Some of the notation (e.g., $\wedge$ represents conjunction, $\implies$ for implies) is standard in the logic, verification and complexity communities, and we will add appropriate citations. Most importantly, we will make notations consistent throughout. This should make it easier to read and parse the proofs (e.g., of Theorem 1).
>
> A2. Limiting to GBDTs: We have answered this in our common response to all since multiple reviewers asked this question.
>
> A3. Connection between sensitivity and p-sensitivity: In sensitivity definition 3.1, we only check if the classifier has changed its decision and the sigmoid function is implicitly present. To produce the decision, a GBDT typically gives a real number as output and we compare the sigmoid of the output against 0.5. According to the definition, a change in the sigmoid output from 0.49 to 0.51 will make a model sensitive, which is not desirable. Therefore in definition 3.2, we unpacked the usage of the sigmoid function and introduced a parameter p that ensures the wide gap between the two outputs. We will clarify this important connection in the revised version of the paper.
>
> A4. Sigmoid function in encoding: Please note that we are very much still considering sigmoid in the encoding, which is why we compute $\delta$ from p at the line 310. More precisely, the function over p  is the inverse function of sigmoid.  Our encoding takes into account the complexity of the sigmoid function, this is the reason for our approximation. We will make this precise and clarify the treatment of the sigmoid function.
>
> A5. "max,min and avg time in experiments": For each benchmark, we check sensitivity against each feature. Then, we report the minimum, maximum, and average running times over all the features of a benchmark. For each feature we only had a single run. But indeed, we could also repeat the runs and take an average over multiple runs; we expect the numbers to not change too much by doing so.

---

> ### Comment · Reviewer_Mrks · 2024-11-21
>
> I acknowledge reading the rebuttal and thank the authors for their answers. I also respect the perspectives of the other reviewers.
>
> Be aware that I have a mathematical background and am very fond of logic and its symbols (and meaning). As such, I find that clarity and structure in mathematical writing are paramount for effective communication. The readability of the arguments could significantly improve with a few adjustments. Some mathematical writings are standard, even if I acknowledge that authors have their own stylistic preferences. For instance, beginning a sentence with a symbol, as in line 211 with $\exists x'$, detracts from clarity and should be avoided. Similarly, line 307 presents the same issue (and others, see below).
>
> Moreover, I provoke the authors to pay closer attention to punctuation, particularly in mathematical contexts (e.g., lines 281 and 307). To mark this point further, Equations (1) to (6) are presented rather abruptly and would benefit from more coherent exposition, enhancing the reader's experience and understanding. A mathematical discourse should flow seamlessly, preventing readers from feeling the need to backtrack frequently.
>
> Another point I must emphasize is the use of logical symbols, such as conjunctions, which are typically interpreted within specific models of some logic (e.g., assignments of truth values to atoms/propositional variables in propositional logic). To clarify matters, the conjunction could have more-than-binary truth values given the algebraic interpretation of the logical symbols/connectives (e.g., many-valued/fuzzy semantics). Without semantics or context for these symbols, their use can lead to confusion, especially for readers not well-versed in the authors' specific approach. For clarity, replacing symbols with their descriptive words (e.g., using "and" instead of $\wedge$ or "exists" instead of $\exists$) could significantly lighten the unnecessary cognitive load for readers.
>
> Please avoid getting overly carried away by subscripts, such as in Equations (1) to (6).
>
> Overall, I recommend looking into Donald Knuth's book/report on "Mathematical Writing" and its references, which could be a constructive resource for developing more precise mathematical argumentation/communication. The report is available at https://jmlr.csail.mit.edu/reviewing-papers/knuth_mathematical_writing.pdf.
>
> While the problem addressed in the paper is intriguing, and I started with enthusiasm to understand the results and proofs, I felt let down by the presentation, like a false promise.
>
> Although I understand the rationale behind the paper's title, I would have preferred a more descriptive approach, such as "Sensitivity Verification for Additive Decision Tree Ensemble" or "Sensitivity Verification for (Additive) Decision Tree Ensemble," which may convey the essence of the work without overstating its significance.
>
> Given these observations, I could review the paper and the generalized results in response to the rebuttal one more time, but I must emphasize that this will require considerable time to ensure the proofs hold up to scrutiny. I find it puzzling, however, that a complete proof was not included in the rebuttal, especially given the effort put into crafting the proof sketch.
>
> I can consider adjusting the assessment score in light of the other reviews, but ultimately, a rejection is warranted at this stage. I hope the authors take this feedback constructively for their future work.

---

> ### Comment · Reviewer_Mrks · 2024-11-28
>
> I have just read the updated version of the paper.
>
> First, I must congratulate the authors on the significant revisions made to the paper. I am happy to say that I understand their contribution and that the paper should be accepted.
>
> Second, I still have some minor observations that would further improve the paper overall:
> - line 100: "(e.g., gradient boosting etc)." --> "(e.g., gradient boosting, etc.)."
> - I assume you use \paragraph{Some title} to list the paragraphs. Specifically, in some cases, "Some title" is written using camel-case, and others are not. For example, "Other related work" versus "Decision Trees", "Decision Tree Ensemble", etc. It is minor but gives a sense of cleanliness.
> - lines 155--156: Please avoid using $\ge$ in textual explanations: use "greater than or equal to". The same holds for $<$ (i.e., write "less than").
> - line 161: use curly braces around $f$, as already pointed out in previous reviews.
> - line 189 and line 194: Avoid using "wrt". Use instead "w.r.t." or (better) "with respect to".
> - Definition 3.2: I would write it better, like:
> > Given a tree ensemble classifier $c : \mathcal X \to \mathcal Y$, a set of sensitive features $F \subseteq \mathcal F$, and a parameter $p \ge 0$, $c$ is said to be $(p,F)$-sensitive if we can find two inputs $x,x' \in \mathcal X$ such that $x_{\mathcal F \setminus F} = x'_{\mathcal F \setminus F}$, $\sigma(c(x)) > 0.5 + p$, and $\sigma(c(x')) < 0.5 + p$. [$\ldots$]
>    - I have added "a set of"
>    - I have removed the unnecessary parentheses around $x$ and $x'$ when defining their projection onto $\mathcal F \setminus F$; for example, $x_{\mathcal F \setminus F}$ instead of $(x)_{\mathcal F \setminus F}$.
>    - I have changed the $\ge$ and $\le$ with $>$ and $<$ in the thresholding part to reflect the encoding at line 325. My view could be wrong on this point, but either way, both the encoding and definition should be coherent with the ordering relations (e.g., both should use $\ge$ or $>$, and not the way it is now in the paper).
> - line 325: "[$\ldots$] as follows." --> "[$\ldots$] as follows" (i.e., without the dot).
> - line 329: "Let $c$ consists of trees $T_1,\ldots,T_m$." --> "Let $c$ be a decision tree ensemble with $m$ trees $T_1,\ldots,T_m$."
> - line 342: "[$\ldots$] ... increasing order." --> "[$\ldots$] ... increasing order as follows"
> - line 345: The symbol $x_1$ should be $x$ and $x_2$ should be $x'$, respectively. Also, "$j \in 1..k_j$" --> "$j \in$ {$1,\ldots, k_j$}".
> - lines 348-349: The sentence "For each node [$\ldots$] evaluating $T_i(x_q)$." is rather clumsy. Perhaps a better way to write it is for example:
> > Let $t_{qin}$ indicate the node $n \in T_i$ visited when evaluating $T_i(x_q)$.
> - line 349: "[$\ldots$] always true." --> "[$\ldots$] always true, encoded as follows". Similar observations hold for lines 353, 357, 366, and 655 (i.e., remove the dots, please).
> - Paragraph "Encoding output": You've changed throughout the paper $x_1$ and $x_2$ into $x$ and $x'$. It seems that this is not the case in this paragraph. Please fix it for better readability.
> - Equation 6: There is a missing $\alpha$ in the right hand side part of the Boolean formula, i.e., it should be $t_{2in}\lfloor \alpha t.val \rfloor$
> - lines 431 and 706: "doesn't" --> "does not" (!)
> - Table 2: I'm confused as to why the entry [IJCNN dataset, 5x] says "0%" when I was expecting "TO" (timeout) since 5x takes longer than 1x.
> - line 470: What is "gap" in "$2 \times $ gap"?
> - line 495: "pseudo-boolean" --> "pseudo-Boolean" (perhaps you should search for other such typos; I am not aware of others).
> - line 660: "[$\ldots$] below:" --> "[$\ldots$] below." (i.e., with the dot)
> - The Claim in the proofs of Theorem 1 and Corollary 2 is the same. I suggest using a Lemma resembling the Claim and using this result (the Lemma) to prove the results. Also, please (again) avoid using $\implies$ instead of the word "implying" in the Claim. **This would be a nice improvement.**
> - lines 712--713: The sentence is very hard to understand. I would suggest something like:
> > [$\ldots$] we can show that a novel yet interesting problem, which we call the differentiating input problem, is also NP-hard.
> - line 722: "[$\ldots$] clause, $cl_i = \ldots$" --> "[$\ldots$] clause $cl_i = \ldots$" (i.e., there is no need for the comma there since it disrupts the reading)
> - Appendix C: Please refer to Tables 4, 5, and 6 in the text in some way.
>
> Finally, this improved paper version celebrated my initial enthusiasm—many thanks to the other reviewers, who ultimately guided the authors in improving their work. This version is now accessible to a general ICLR audience, and the reading flows elegantly amidst intuitions, motivations, and mathematical arguments. I hope the authors will take the time to implement my last observations. As such, I'm increasing my score from 3 to 6.

---

> > ### Author Response · Authors · 2024-11-29
> >
> > Thanks so much for the in-depth reading and feedback. We will try to implement these last observations carefully.

---

### Official Review · Reviewer_1wTC · 2024-11-04

**Soundness:** 4
**Presentation:** 3
**Contribution:** 2
**Rating:** 6
**Confidence:** 3

**Summary:**

The paper puts forward an approach for formal verification of feature sensitivity. Feature sensitivity is a characteristic of a model whereby the model changes its output depending on values of such features. In domains such as fairness checking for feature sensitivity is one of the ways in which fairness checks are meaningfully encoded.

The paper answers some complexity problems on answering features sensitivity, leaves some open. Crucially it provides a novel and efficient way based on pseudo-Boolean encodings to solve the verification problem. Some experimental results showing SoA performance are provided.

Overall this is a solid paper, arguably with some weaknesses, that advances the SoA of the area.

Post-rebuttal note: The authors answered by questions and provided additional material (developed post-submission) to solve some of the questions left unanswered in the paper. On the other hand I also noticed some presentation issues that I think can be fixed. I think the final answer on scalability on the point are raised remains unresolved; so it is still not 100% clear to me the extent to which the novel encoding is responsible for the gain. I do accept the authors point in as far as the comparison against SMT goes.
Overall I am still mildly positive on the paper.

**Strengths:**

The paper is very well presented (some very minor typos that can be fixed). I cannot judge the soundness of the encoding but it seems reasonable. I read the complexity proofs and appear correct (even if not surprising). In terms of advancement on SoA I judge this to be in line with the standards expected at ICLR, even if perhaps not as a top paper (see weaknesses below).

Most importantly solving model valuation for ensembles remains a key question in a variety of domains and this paper presents a principled and noteworthy contribution to the challenge. While the NP-hardness results may not be surprising they fill a gap in the knowledge in the area and the encoding presented achieves SoA performance.

**Weaknesses:**

While the NP hardness won't be a surprise to most, not all the theoretical questions are answered in the paper with the obvious gap being trees of depth 2 and 3. I feel this has a considerable implication even in terms of architectural suggestions from this study. Has this question been answered by the authors in the meantime? To me these corner cases are actually the most interesting ones. Finding the the case for 3 is polynomial would be a very interesting result

The reasons as to whether the approach appears to scale considerably better than present SoA were not clear to me in the paper. Do the authors have an explanation? If so, could they provide it? The reason for asking is that in principle it could even be that their implementation is just more efficient and it does not have so much to do with the encoding.

**Questions:**

See questions on the weaknesses above.

---

> ### Author Response · Authors · 2024-11-18
> **Pseudo-Boolean encoding vs other SOTA approaches**
>
> Thanks for the very positive comments and remarks, as well as the intriguing and to-the-point questions. Regarding NP-hardness, we were also hoping for polynomial time algorithms for depth 2/3. Unfortunately, for the case of depth 3 we have since managed to prove that the problem of 1-feature sensitivity continues to be NP-hard. We have outlined this argument in the common response, as multiple reviewers had asked for it. But to add to it, we believe that while there is still an obvious theoretical gap, this paper represents a substantial improvement over the state-of-the-art, in the problem formulation as well as our encoding. In most practical benchmarks that we saw, the trees were of depth at least 3, and often of depth 5 or 6. Further, our answer below should shed more light on why our encoding is a main contribution and shows fundamental novelty (not just efficient implementation, though we agree and hope that this is also a reason for our good performance!).
>
> --------
> This bring us to the rather interesting question of why we are performing better, due to better engineering or better encoding. Our first remark is that pseudo-Boolean solvers employ a combination of linear arithmetic and Conflict-driven clause learning (CDCL for short) that are a natural fit for problems like knapsack problems. Given our sensitivity problem is similar to the knapsack problem, we feel that pseudo-Boolean constraints are a somewhat natural encoding for our problem. In the field of SAT solving, pseudo-boolean solving is being actively studied. Consequently, we also expect the solver developers will be greatly interested in improving the performance of their solvers on the sensitivity problem.
>
> The earlier approaches used completely different encodings: Veritas[1] uses a very different method for the problem, which is a custom implementation on k-cliques over output configurations to solve the sensitivity problem. Essentially, Veritas is a custom-made solver for an NP-hard problem. It is usually very hard to design such solvers and compete against state-of-the-art solvers for equivalent problems. There are cases where dedicated solvers outperform generalized solvers, e.g., graph isomorphism [2]. However, the best tools in the area are the result of the effort of a lifetime. Therefore, we believe that the best approach for such problems is to encode them into one of the well-studied problems, and hope the solvers will improve over time to solve the encoded instances.
>
> Note further that, as mentioned in the paper, our encoding of the problem differs significantly from the one presented in [3] Ignatiev et al. (2020a), which is a direct SMT encoding of trees and their verification query. Encoding the problem as SMT constraints, as done in [3], may result in a loss of structural information, preventing the solver from fully leveraging the pseudo-Boolean nature of the constraints. We solve these constraints using a pseudo-Boolean solver to check for p-sensitivity.
>
> In summary, we believe that our advantage in performance is really due to a combination of novelty in choice of encoding combined of course with an efficient implementation, which can directly take advantage of the advances in state of the art pseudo-Boolean solvers. We hope that somewhat answers your  question.
>
> [1]  Veritas: Versatile Verification of Tree Ensembles. Laurens Devos, Wannes Meert, and Jesse Davis. ICML 2021 http://proceedings.mlr.press/v139/devos21a.html
>
> [2] https://github.com/ciaranm/glasgow-subgraph-solver
>
> [3] Ignatiev, A., Cooper, M.C., Siala, M., Hebrard, E., Marques-Silva, J. (2020). Towards Formal Fairness in Machine Learning. In: Simonis, H. (eds) Principles and Practice of Constraint Programming. CP 2020. Lecture Notes in Computer Science(), vol 12333. Springer, Cham. https://doi.org/10.1007/978-3-030-58475-7_49

---

> > ### Author Response · Authors · 2024-11-27
> >
> > We hope that our responses above, especially with regards to (i) NP-hardness at depth 3 and (ii) reasons for scalability compared to SoTA, were sufficient to allay the reviewer’s concerns. We will incorporate both of these into the revised version.

---

### Official Review · Reviewer_ULi2 · 2024-11-05

**Soundness:** 3
**Presentation:** 4
**Contribution:** 3
**Rating:** 8
**Confidence:** 3

**Summary:**

This submission considers the problem of sensitivity for tree ensemble classifiers, i.e. the question of determining if, for a selection \\(F\\) of features, there exists a pair of inputs that agree on all the features outside of \\(F\\) but get classified differently by the tree ensemble. This work also develops a quantitative version of sensitivity (Def 3.2), parametrised by a number \\(p\in[0,\frac{1}{2}]\\), that additionally lower-bounds the "distance" between the two classifications.

The main theoretical contributions consist in proving that
- the (non-quantitative) sensitivity problem is NP-hard (Thm 1)
- the sensitivity problem restricted to singleton feature sets F={f} is also NP-hard (Thm 2)

The main practical contribution is to show that the problem of establishing \\(p\\)-sensitivity can be encoded as the satisfiability of a system of pseudo-boolean constraints (Thm 3). Experimental evidence shows that this approach scales much better than SOTA methods.

**Strengths:**

The paper is very well-written, with a good overview of the literature and easy-to-follow examples, proofs and discussions. In particular the proofs are easily understandable even by someone with very limited knowledge of computational complexity theory such as myself.

The main practical novelty of this paper -- using a pseudo-boolean satisfiability encoding -- is credibly shown to be superior to the SOTA methods in a short but convincing experimental section.

**Weaknesses:**

Theorems 1 and 2 correspond to two extremes of the sensitivity problem: the first considers sensitivity w.r.t. the full set of features, the second considers only singleton subsets of features. It feels like the full story would be a result showing that sensitivity w.r.t. subsets of features of size \\(N\\) is NP-hard, for all \\\(N\\). Similarly, the story is slightly incomplete as in does not cover trees of depth 2 or 3 (as mentioned by the author(s)). The paper would be nicer (and stronger) if these questions were solved. too

Here are some minor comments/corrections
* l69: when THE number
* l83: whether...whether
* l104: some OF these
* l117: of A \\(d\\)-dimensional
* l157: When \\(F=\\{f\\}\\)
* l168: two-tree
* l194: Def 3.2 is not really a special case of Def 3.1 in the sense that Def 3.1 does not correspond to a particular choice of \\(p\\). As it stands, Def 3.1 is equivalent to \\(\exists p.\\)Def 3.2 holds. To make Def 3.1 a special case of Def 3.2 I would change it to have \\(p\geq 0\\) and \\(\sigma(c(x_1))\geq 0.5+p, \sigma(c(x_2))< 0.5+p\\) (or the other way round). Then Def 3.1 would correspond to the case \\(p=0\\).
* In the proofs of thm 1 and thm 2, why not simply take \\(X_{k+1}=\\{-1,1\\}\\) and \\(X_0=\\{-1,1\\}\\)?
* l275: \\(F\cup\\{f\\}\\) (not \bigcup and singleton)
* l289: I don't understand the first sentence here. Which problem requires three variables \\(x_1,x_2,x\\)?
* l307 and l313: use \setminus for the set difference \\(\mathcal{F}\setminus F\\). Also, if you'd used \\(x,x'\\) or \\(x,y\\) instead of \\(x_1,x_2\\), you wouldn't have problems with double subscripts.
* l310: \log instead of log
* eq (2): shouldn't \\(j+1\\) in the RHS simply be \\(j\\)?
* l327-328: Incomplete sentence
* l328: Since THE root
* Thm 3: use \eqref in the conjunction, in order to get \\((1)\wedge\ldots\wedge(6)\\). Parentheses = equations. Conversely don't use (1) and (2) to refer to the two parts of the theorem, instead use 1. and 2., or better (i) and (ii).
* l394: mine -> us
* l420: p-sensitive -> \\(p\\)-sensitive

**Questions:**

Do you think it is the case that sensitivity w.r.t. subsets of features of size \\(N\\) is NP-hard, for all \\\(N\\)? If yes, do you have any intuition about whether proving this is substantially harder than your existing proofs?

Same question for trees of depth 2 or 3.

---

> ### Author Response · Authors · 2024-11-18
> **Sensitivity for N-sized subsets and other questions**
>
> Thanks for the positive comments and nice observations (and suggestions, which we will certainly incorporate). To address the main question,  we believe that the sensitivity problem with respect to subsets of any fixed size is also NP-hard. One way to see this is, given an instance of 1-feature sensitivity, we can construct an instance of a $k$-subset sensitivity problem by introducing $k-1$ new dummy variables to the problem. These dummy variables are part of our input to sensitivity checking but they don't affect the tree's output in any way (formally, a possible way to add these is to have decision tree stumps on these variables that output 0 irrespective of the variables values). Thus, checking for sensitivity for $k$-subset of features will be equivalent to checking for sensitivity for just the first feature in the original instance and thus checking sensitivity with respect to a given subset of features is also NP-hard. We will add this as a corollary.
>
> Regarding the second main question, i.e., for the case of depth 2 and 3 trees, please see the common response, where we have sketched a proof of NP-hardness for depth 3. Unfortunately, we don’t have an answer for depth 2 presently (the proofs in this paper will not directly solve this case), but we hope that this paper will lead to further work towards this too.
>
> In addition to the above main questions, there were minor remarks, of which we address three main questions that occurred in them; we will carefully fix these and the rest.
> First, we agree that in Def 3.2 we need to write $p\geq 0$ to make it subsume 3.1, we will fix this.
> Second, regarding the question of why we can’t take X=\{-1,1\} in theorem 1 and 2 instead of reals, we agree that the proof only requires a separation above and below 0. We felt that keeping reals throughout would make the proof homogeneous, but we agree that we could also just do as suggested.
> Finally, in l289: we apologize for a typo here: the third $x$ should have been $x_{\bot F}$. More precisely, we mean that if we take any candidate solution, i.e., as assignment $x_1 \in \{0,1\}^{|F|}, x_2 \in \{0,1\}^{|F|}$ and $x \in \{0,1\}^{|\mathcal{F}\setminus F|}$, we can then run the decision trees and aggregate the answer to check if it is valid.

---

> > ### Comment · Reviewer_ULi2 · 2024-11-26
> > **Improvements made**
> >
> > I'm impressed by the improvements made to this submission. Almost all my questions have been addressed. If all the new results are written up and integrated clearly in the submission (and I think they will), then this will be a very satisfying paper in that virtually no outstanding questions will remain on this specific topic. The only case that still isn't covered, the depth 2 case, isn't covered for a reason that the author(s) explained well and can therefore be left open for now.

---

> > > ### Author Response · Authors · 2024-11-27
> > >
> > > Thank you so much for the positive feedback. Indeed, we are in the process of integrating the new results (and fixing the typos and minor corrections pointed out) and will upload the revised version by the deadline.

---

### Official Review · Reviewer_joG2 · 2024-11-10

**Soundness:** 2
**Presentation:** 3
**Contribution:** 2
**Rating:** 6
**Confidence:** 3

**Summary:**

This paper addresses the feature sensitivity problem for ensembles of decision trees, with a focus on Gradient Boosted Decision Trees (GBDT). The authors demonstrate that this problem is NP-complete through a reduction from the SAT problem. They then propose an algorithm to solve the p-sensitivity problem by encoding it as pseudo-Boolean constraints. Finally, experimental results are provided, comparing the proposed approach to some existing methods.

**Strengths:**

The paper is well-written, with a clear structure that makes it easy to follow. This clarity enhances the readability and understanding of the material.

The problem is well motivated.

The theoretical contributions are interesting and their proofs appear to be correct.

**Weaknesses:**

The initial result on NP-hardness of sensitivity problem for all features seems redundant after establishing NP-completeness for a single feature.

My main issue with this paper is with the experiments. The comparison with VERITAS seems a bit unfair. By limiting VERITAS to the same runtime as SENSPB, the comparison may not reflect its full potential. Allowing VERITAS more runtime might yield better results, or it could have already produced sufficiently good results within its runtime. A separate table detailing the number of instances each method solved and the time taken would provide additional insights.

**Minor Comments/Typos**
- Line 068: "just on" → "just one"
- Line 079: Rewrite this sentence for clarity: “Can we use other forms of powerful reasoning, Pseudo-Boolean solvers, that have shown to be effective in other problems (Mexi et al., 2023), for the sensitivity problem?”
- Line 162: "given set of sensitive features F" → "given set of features F"
- Caption of Table 1: "results than mine" → "results than our method"

**Questions:**

Is there a reason why the NP-hardness of the all feature problem was included in the paper? If it can be useful in some cases, some explaination here can be beneficial.

It would be nice to know why the results are specific to GBDT and why they could/couldn't be directly extended to other ensemble models, such as random forests.

---

> ### Author Response · Authors · 2024-11-18
> **Veritas Experimental Comparison and other responses**
>
> Regarding the comment on comparison with Veritas, we note that Veritas is an anytime stoppable tool. In the paper, when we say we limit Veritas' runtime, we stop it and look at the results found by VERITAS during that runtime and compare it with the bounds found by SensPB. At the same time, we agree that it would be interesting to look at what happens if Veritas is run for a longer time. We considered the following two experiments:
> 1) If we fix a uniform timeout of 3600s, how does the performance of Veritas compare with SensPB, i.e., how long does it take Veritas to reach an optimal solution ? However,  the two tools solve different problems, and comparing them using this metric is not the best solution. SensPB stops execution when we find one counterexample, while Veritas tries to find inputs that produce the maximum possible difference between the outputs and is thus doing more heavy work. We thought it would be interesting to see how the quality of the results produced by Veritas changes when we let it run for longer, but not till an ad-hoc timeout. Hence we also considered the following question.
> 2) If given, say, 5 times the runtime of SensPB, what is the quality of solutions produced by Veritas?
>
> Below we give the results of our experiments:
> 1) Running Veritas with timeout = 3600 s
>
> +---------------------------+------------------+------------------+
>
> | Benchmark Name            | Time SensPB (s) | Time Veritas (s) |
>
> +---------------------------+------------------+------------------+
>
> | breast cancer robust        | 2.7              | 2.5               |
>
> | breast cancer unrobust    | 2.7              | 2.53             |
>
> | diabetes robust                | 3.0              | 3.2               |
>
> | diabetes unrobust            | 5.9              | 198.1           |
>
> | cod-rna unrobust             | 8.3              | 346.2            |
>
> | Binary-mnist robust         | 14.9             | TO               |
>
> | higgs unrobust                | 1188             | TO              |
>
> | IJCNN unrobust              | 330              | OOM           |
>
> +---------------------------+------------------+------------------+
>
> *TO = Timeout, OOM = out of memory
> For Binary MNIST robust and Higgs unrobust, Veritas always times out, without producing a single solution. For IJCNN robust, Veritas runs out of memory after roughly 900s.
>
> 2) 5*runtime experiment
>
> Let the time taken by SensPb be t. We run Veritas for sensitivity analysis, one feature at a time, with the following timeouts: 2t, 5t and 3600 (the default timeout). We look at the bound produced by Veritas at the end of the timeout and compare this bound to the bound found by SensPb. In the following table we report the fraction of features in which Veritas performed better than or equivalent to SensPb:
>
> +---------------------------+-------+-------+-------+
>
> | Benchmark Name           |  2x   |  5x   | 3600  |
>
> +---------------------------+-------+-------+-------+
>
> | breast cancer robust       | 81.82 | 81.82 | 81.82 |
>
> | breast cancer unrobust    | 45.45 | 45.45 | 45.45 |
>
> | diabetes robust                 | 77.78 | 77.78 | 77.78 |
>
> | diabetes unrobust           | 66.67 | 77.78 | 88.89 |
>
> | cod-rna unrobust             |  0.00 | 25.00 | 37.50 |
>
> | Binary-mnist robust         | TO     |    TO   |   TO    |
>
> | higgs unrobust                | TO     |    TO    | TO    |
>
> | IJCNN unrobust              |  TO     |     0   | OOM     |
>
> +---------------------------+-------+-------+-------+
>
> As expected, on increasing the time given to veritas, it starts performing better on more and more features (specifically, look at diabetes unrobust and cod-rna unrobust). However, even on running Veritas for 3600s it leaves out a lot of features in which SensPb performs better. As noted earlier, for three of our largest benchmarks, Veritas does not produce a single solution in the time given.
>
> --------
>
> Regarding remaining comments, first, we would like to thank the reviewer for pointing out several typos, we will fix all of them in the final version. Also, in response to the questions, we decided to include the NP-hardness of the all feature problem mainly for pedagogical reasons. However, especially in light of the stronger depth-3 result, we could perhaps just focus on the single feature case, and explain (as a corollary) how it generalizes to subset/all features rather than write it separately. Finally, in response to the last question, indeed, our results (esp theoretical and encoding) are not specific to GBDTs and can be lifted to any tree ensemble model (including random forests) as long as we sum up the decisions as individual trees. As this question was asked by multiple reviewers, we have provided the details in the common response above.

---

> > ### Author Response · Authors · 2024-11-27
> >
> > We hope our response regarding the main weakness identified by the reviewer, i.e., experimental comparison with VERITAS, is adequate. To summarize, we have run the additional experiments asked for (and few more that we felt useful) and presented the results in our response above. We plan to add these results (within space limits) to the revised manuscript that we will upload by the deadline.

---

> > > ### Comment · Reviewer_joG2 · 2024-11-27
> > >
> > > I thank the authors for addressing the concers raised by me and the other reviewers. The new experiments show that their approach is better on some benchmarks and comparable on others. I would still suggest to show the two colums "% of features on which SensPb performs strictly better" and "% of features on which Veritas was strictly better". With these results included in the paper along with the NP-hardness of depth 3 case while removing the NP hardness of all feature case, I would be happy to accept this paper. However, this seems like a major revision.

---

### Author Response · Authors · 2024-11-18
**Common response to all reviewers regarding random forests**

We thank all the reviewers for their careful reading and useful feedback, comments, and questions. Here we address a common question posed by multiple reviewers so as to not duplicate our responses later, namely, whether our approach is specific to GBDT models or works for general tree ensembles. Our response is four-fold:

- Our definitions of the sensitivity problem indeed hold for general tree ensembles.
- Our hardness results also apply to tree ensembles. Note that in Section 3, we only use that it is a decision tree ensemble classifier. The only extra assumption we need is that the classifier “ensembles” the trees by summing up the decisions of individual trees (this is sometimes called additive tree ensembles , see e.g.,[1]). In particular, our results are independent of the way the tree ensemble was trained (e.g, gradient boosting etc).
- Our encoding in Section 4 also directly works for any tree ensemble which aggregates the trees by summing up the individual tree decisions.
- In our experiments, we focused on GBDT models because they are a large and well-known class that have (i) easy to access benchmarks and (ii) are additive.

In summary, we can write/extend our definitions, theoretical results as well as our encoding to random forests if they follow an additive prediction model.

A caveat is that the classical definition of random forests [2] computes the final answer by max-pooling rather than summing up, which we do not handle in this paper. We believe that our theoretical results and encodings can also be extended to the max-pool setting, but we leave this for future work. On the other hand, some famous implementations, such as Scikit-Learn [3], do use a weighted average of the individual tree predictions to make the final decisions, which can be incorporated into our approach by understanding and incorporating their weights. We did not run experiments for (additive) Random Forests separately, since from the point of view of our encoding, they are equivalent to GBDT models.

We hope this answer also justifies the title of the paper and the focus on (additive) tree ensembles. We will edit our presentation in the paper (e.g., in the introduction) to rewrite and clarify these points.

[1] Veritas: Versatile Verification of Tree Ensembles. Laurens Devos, Wannes Meert, and Jesse Davis. ICML 2021 http://proceedings.mlr.press/v139/devos21a.html

[2] Random Forests. Breiman, L. Machine Learning 45, 5–32 (2001). https://doi.org/10.1023/A:1010933404324

[3] Scikit-Learn https://github.com/scikit-learn/scikit-learn/blob/6e9039160f0dfc3153643143af4cfdca941d2045/sklearn/ensemble/_forest.py#L883

---

### Author Response · Authors · 2024-11-18
**Common response regarding depth 3 hardness**

A second common question of multiple reviewers is whether our results could be extended to decision tree ensembles of depth 2 and 3. Indeed, our current hardness results kick in only for decision tree ensembles of depth at least 4. But since the submission deadline, we have been able to modify the existing proofs to prove NP-hardness for depth 3 decision trees, a strictly stronger result than what is currently in the paper. We will add/replace it in the revised version of the paper, if permitted (and add hardness of distinguishing input as a corollary). This still leaves the case of depth 2 trees open. None of the present (including the new)  proofs can be extended to this case, since both our hardness reductions and results using evasiveness rely on hardness of 3-CNFSAT, which requires 3 variables per clause which we translate to depth 3 trees. However, 2-CNFSAT is poly-time solvable and hence not useful to show hardness. We expect that a different technique/reduction/encoding will be needed to resolve this question.

We sketch the depth 3 proof below:
----------
We reduce from 3CNF-SAT which asks given a Boolean formula in conjunctive normal form (CNF) with 3 variables per clause, whether it is satisfiable. Given an instance $\varphi$ of 3CNF-SAT, let $cl(\varphi)$ be the set of clauses $\{cl_1,cl_2,\dots,cl_m\}$, with $m=|cl(\varphi)|$ and let $var(\varphi)$ denote the set of variables \{$v_1,v_2,\dots,v_n$\}, with $n=|var(\varphi)|$. Then from $\varphi$ we start by creating the formula $\varphi' = \varphi \land (v_{n+1} \lor v_{n+1} \lor v_{n+1}) $ which is also a 3CNF formula with a new variable $v_{n+1}$ and a new clause $cl_{m+1} = (v_{n+1} \lor v_{n+1} \lor v_{n+1})$. Observe that $\varphi$ is satisfiable, i.e., there exists an input $x\in \{{0,1\}}^n$ that satisfies $\varphi$ iff there exists an input $x'\in \{0,1\}^{n+1}$ that satisfies $\varphi'$. We the construct a decision tree ensemble $d$ with depth 3, such that $d$ is 1-feature sensitive iff $\varphi'$ is satisfiable.

In formula $\varphi'$, for every clause $cl_i$, we create a depth-3 decision tree $T_i$ as depicted in Figure 3 from the submitted paper, where $m=|cl(\varphi)|$ is replaced by $m+1=|cl(\varphi')|$ (we replace the notation of variable $x_i$ by $v_i$ to avoid confusion). That is, for each literal (i.e., $v_i$ or $\neg v_i$) in the clause, we add a "true" branch with output $\frac{1}{|cl(\varphi')|}$, and a "false" branch where we either continue to next literal or return $-1$. For each literal, if it occurs positively as $v_i$ (resp. negatively as $\neg v_i)$, the true (resp. false) branch outputs $\frac{1}{|cl(\varphi')|}$. We form the decision tree ensemble $d$ using the above decision trees with trees enumerated $T_i$ for $i \in \{1,2,\dots, m+1=|cl(\varphi')|\}$. The domain of $d$, i.e., $\mathcal{X}={0,1}^{n+1}$ and $\mathcal{Y}=\{0,1\}$, where depending on whether the sum of values at the leaves of the trees is greater than 0 or not, we output 1 or 0.

Claim A: For all $x\in\{0,1\}^{n+1}$ we have $d(x)=1$ iff  $\varphi'(x)=1$, i.e., $x$ satisfies/models $\varphi'$. To see the proof of this claim, there are two cases: First, suppose the input satisfies the 3CNF formula $\varphi'$, i.e., $\varphi'(x)=1$. In this case, each of the $m+1$ clauses are satisfied in the input and thus, for all trees $T_i$ we have $T_i(x) = \frac{1}{m+1}$. Thus, $\Sigma_1^{m+1} T_i(x) = 1 > 0 \implies d(x) = 1$. Second, suppose the input doesn't satisfy the 3CNF formula $\varphi'$. Thus, there exists a clause which is not satisfied by the input. Let that clause be $cl_j$. By the construction of $d$, for the corresponding tree $T_j(x) = -1$ and $T_i(x) \leq \frac{1}{m+1}$ for all $i \neq j$. Thus, $\Sigma_1^{m+1} T_i(x) \leq -1 + \frac{m}{m+1} = \frac{-1}{m+1} < 0$ which implies that $d(x) = 0$.

Finally, let $F$ be the set containing the singleton ${n+1}^{th}$ feature corresponding to $v_{n+1}$. To complete the proof, we will show that $d$ is $F$-sensitive iff $\varphi'$ is satisfiable. If $d$ is $F$-sensitive, there exist two inputs where the answer is different, from which we infer that there exists $x$ s.t $d(x)=1$ which by above Claim A means $\varphi'$ is satisfiable. The other direction when $d$ is not $F$-sensitive, by considering $x_F=0$, we get $T_{m+1}$ will evaluate to $-1$ since $cl_{m+1}[v_{n+1}\mapsto 0]=0$. As a result, we can conclude that for any $x_{\bot F}\in \{0,1\}^n$, we have $\sum_1^{m+1}T_i(x_{\bot F}, 0) \leq -1+ \frac{m}{m+1}< 0$ and so $d(x_{\bot F},0)=0$. Thus, for any $x_F\in \{0,1\}$, since $d$ is not $F$-sensitive, we get $d(x_{\bot F},x_F)=0$, which implies that for all $x\in\{0,1\}^{n+1}$, $d(x)=0$. Again appealing to Claim A above, we can conclude that $\varphi'$ is not satisfiable, which completes the proof.

---

### Author Response · Authors · 2024-11-28
**Highlighted rebuttal revision**

We have uploaded a highlighted version of our paper, which has all the main changes colored blue for your easy reference. We have tried to do all and only the changes that we promised in our response. We have also tried to fix the typos etc to the best of our ability (these minor changes are not highlighted).

---

### Meta-Review · Area_Chair_MRhy · 2024-12-19

**Metareview:**

This paper considers the feature sensitivity problem for ensembles of decision trees in a verification context.

The reviewers generally agree that there is sufficient novelty in using a pseudo-boolean satisfiability encoding. The experiments indicate that this method is often superior to the state-of-the-art.

Moreover, several weaknesses regarding presentation and clarity have been mitigated in the rebuttal phase. In particular, the experiments were improved, and hence, we recommend acceptance of the paper.

**Additional Comments On Reviewer Discussion:**

All reviewers agree that, especially after the thorough answers and improvement of the paper by the authors, it is worthwhile to accept.

---

### Decision · Program_Chairs · 2025-01-22

Accept (Poster)